# Emissions of halocarbons from India inferred through atmospheric measurements

Daniel Say[1], Anita L. Ganesan[2], Mark F. Lunt[3], Matthew Rigby[1], Simon O'Doherty[1], Christina Harth[4], Alistair J. Manning[5], Paul B. Krummel[6], and Stephane Bauguitte[7]

[1]School of Chemistry, University of Bristol, Bristol, BS8 1TS, UK
[2]School of Geographical Sciences, University of Bristol, Bristol BS8 1SS, UK
[3]School of Geosciences, University of Edinburgh, Edinburgh, EH9 3JW, UK
[4]Scripps Institution of Oceanography, University of California, San Diego, La Jolla, USA
[5]Met Office Hadley Centre, Exeter, EX1 3PB, UK
[6]Climate Science Centre, CSIRO Oceans and Atmosphere, Aspendale, Australia
[7]Facility for Airborne Atmospheric Measurements, Cranfield University, MK43 0AL, UK

**Correspondence:** Daniel Say (Dan.Say@bristol.ac.uk)

**Abstract.** As the second most populous country and third fastest growing economy, India has emerged as a global economic power. As such, its emissions of greenhouse and ozone-depleting gases are of global significance. However, unlike neighbouring China, the Indian sub-continent is very poorly monitored by existing atmospheric measurement networks. India's halocarbon emissions (here defined as chlorofluorocarbons (CFCs), hydrochlorofluorocarbons (HCFCs), hydrofluorocarbons (HFCs) and chlorocarbons) are not well-known. Previous measurements from the region have been obtained at observatories many hundreds of kilometres from source regions, or at high altitudes, limiting their value for the estimation of regional emission rates. Given the projected rapid growth in demand for refrigerants and solvents in India, emission estimates of these halocarbons are urgently needed to provide a benchmark against which future changes can be evaluated. In this study, we report the first atmospheric-measurement derived halocarbon emissions from India. Air samples were collected at low-altitude during an aircraft campaign in June and July 2016 and emissions were derived from measurements of these samples using an inverse modelling framework. These results were evaluated to assess India's progress in phasing out ozone-depleting substances under the Montreal Protocol. Our combined CFC estimates show that India contributed 54 (27 – 86) Tg $CO_2$eq yr$^{-1}$, and HCFC-22 emissions at 7.8 (6.0 – 9.9) Gg yr$^{-1}$ were of similar magnitude to emissions of HFC-134a (8.2 (6.1 – 10.7) Gg yr$^{-1}$). We estimate India's HFC-23 emissions to be 1.2 (0.9 – 1.5) Gg yr$^{-1}$ and our results are consistent with resumed venting of HFC-23 by HCFC-22 manufacturers following the discontinuation of funding for abatement under the Clean Development Mechanism. We report small emissions of HFC-32 and HFC-143a and provide evidence to suggest that HFC-32 emissions were primarily due to fugitive emissions during manufacturing processes. Lack of significant correlation among HFC species and the small emissions derived for HFC-32 and HFC-143a indicate that in 2016, India's use of refrigerant blends R-410A, R-404A and R-507A was limited, despite extensive consumption elsewhere in the world. We also estimate emissions of the regulated chlorocarbons carbon tetrachloride and methyl chloroform from Northern and Central India to be 2.3 (1.5 - 3.4) Gg yr$^{-1}$ and 0.07 (0.04 - 0.10) Gg yr$^{-1}$ respectively. While the Montreal Protocol has been successful in reducing emissions of many ozone-depleting substances, growth in the global emission rates of the unregulated very short-lived substances poses an

ongoing threat to the recovery of the ozone layer. Emissions of dichloromethane are found to be 96.5 (77.8 - 115.6)Gg yr$^{-1}$ and our estimate suggests a 5-fold increase in emissions since the last estimate derived from atmospheric data in 2008. We estimate perchloroethene emissions from India and chloroform emissions from Northern-Central India to be 2.9 (2.5 - 3.3) Gg yr$^{-1}$ and 32.2 (28.3 – 37.1) Gg yr$^{-1}$ respectively.

# 1   Introduction

Chlorofluorocarbons (CFCs), carbon tetrachloride (CTC) and methyl chloroform (MCF) were used widely around the world for refrigeration, air-conditioning, foam blowing and solvent applications (Montzka et al., 1999), until they were found to deplete stratospheric ozone (Molina and Rowland, 1974; Engel et al., 2019). These species were thus regulated under the
Montreal Protocol on Substances that Deplete the Ozone Layer. The adoption of the Montreal Protocol and its amendments subsequently led to a marked reduction in emissions of ozone-depleting substances (ODSs) (Engel et al., 2019). However, emissions are expected to continue, particularly from developing (Article 5) nations (Vollmer et al., 2009; Wan et al., 2009), predominantly from banked sources such as refrigerators and rigid foams (Vollmer et al., 2009), and as fugitive emissions from industry (Sherry et al., 2018).

While the emissions of many ODSs are declining, emissions of some are at odds with expectations. Emissions of CFC-11 have been shown to be increasing since 2013 (Montzka et al., 2018) and evidence suggests new production from China which has not been reported to the United Nations Environment Programme (UNEP) (Rigby et al. (2019), in press). While 'bottom-up' emissions of CTC, estimated from reported consumption for feedstock use, are small (1 – 4 Gg yr$^{-1}$ (Montzka et al., 2011)), 'top-down' studies, based on atmospheric observations, suggest actual global emissions still exceed 30 Gg yr$^{-1}$
(Chipperfield et al., 2016; Liang et al., 2014; Lunt et al., 2018).

As a consequence of the Montreal Protocol, emissions of the first-generation of CFC replacements, the hydrochlorofluorocarbons (HCFCs), species with similar thermodynamic properties to CFCs but reduced ozone-depletion potentials (ODPs), increased considerably in the 1990s and 2000s (Montzka et al., 2009). Because HCFCs still have non-zero ODPs, they were subsequently also regulated under the Copenhagen Amendment to the Montreal Protocol in 2004. Article 5 countries are still
permitted to emit HCFCs but began their HCFC phase-out in 2013, with reduction targets outlined by the HCFC phase-out management plan (HPMP (UNDP, 2013)). Recently, it was reported that global emissions of the three major HCFCs (HCFC-22, HCFC-141b and HCFC-142b) had stabilised or were decreasing, largely due to decreasing emissions from the developed world (Montzka et al., 2014; Simmonds et al., 2017).

Following regulation of HCFCs, the second-generation replacements, hydrofluorocarbons (HFCs) were adopted because
they do not appreciably deplete stratospheric ozone. However, their high Global Warming Potentials (GWP, Table 1) mean that HFCs contribute to global climate change, leading to recent efforts to reduce consumption. The 2016 Kigali Amendment to the

Montreal Protocol set out targets for a gradual phase-down of HFC production and consumption. The first cuts by most Article 5 countries will not be required until 2024 and a small number of these countries will not be required to freeze emissions until 2028. Except for HFC-23 (Simmonds et al., 2018), whose emissions are a by-product of HCFC-22 production, and HFC-152a, whose emissions have stabilised since 2010 (Simmonds et al., 2016), global emissions of all major HFCs were rising until at least the end of 2016 (Simmonds et al., 2017).

While CTC and MCF are now regulated under the Montreal Protocol, very short-lived substances, such as the chlorocarbons dichloromethane (DCM), perchloroethene (PCE) and chloroform, were not considered a threat to stratospheric due to their short atmospheric lifetimes (Table 1) and thus are not regulated. However, recent studies have shown that the rapid recent growth in global emissions of DCM and chloroform has the potential to delay the recovery of the Antarctic ozone-hole (Hossaini et al., 2017; Fang et al., 2018). Hossaini et al. (2017) estimated global DCM emissions to be ~0.6 Tg yr$^{-1}$ in 2004, which rose to over 1.1 Tg yr$^{-1}$ by 2014. If this trend continues, Hossaini et al. (2017) shows that DCM emissions alone could lead to a delay in the recovery of the Antarctic ozone-hole by 17 - 30 years. Likewise, Fang et al. (2018) estimates that continued growth in global emissions of chloroform could result in a further delay in ozone layer recovery of 6 – 11 years.

In certain regions of the world, large-scale convective systems provide an efficient route for the transport of short-lived chlorocarbons to the stratosphere without being substantially removed in the troposphere. South Asia's monsoon systems, similar to those over eastern Asia, provide one such pathway (Fadnavis et al., 2013; Randel et al., 2010). Brioude et al. (2010) show that short-lived chlorocarbons emitted from South Asia have ODPs up to 8 times greater than those emitted from elsewhere in Asia, and 22 times greater than emissions from Europe.

India, an Article 5 country under the Montreal Protocol, ratified the Protocol in 1992. A complete phase-out of CFCs, CTC and MCF was mandated in India by 2010. Except for use in metered dose inhalers, which ceased in 2010, India reported a complete phase-out of both the production and consumption of CFCs in 2008 (UNDP, 2013). Emissions from existing banks, such as old refrigeration appliances, are however, likely to persist. Following phase-out of CFCs, India was required to reduce emissions of HCFCs. Under Stage I of the HPMP, production and consumption of HCFCs for dispersive use was designated to be frozen by January 1st, 2016, followed by complete phase-out by 2040. At the 19th Meeting of the Parties in 2007, India agreed to an acceleration of this schedule. Under Stage II of the HPMP, India agreed to freeze its consumption of HCFCs at the base level (2009/10 average) by 2013, followed by a 10% reduction (relative to the base level) by 2015 and a complete phase-out by 2030. In 2016, India adopted the Kigali Amendment, under which it will also begin to phase-down its production and consumption of HFCs. However, its developing status means it will not be required to make its first reductions until 2028, and in the meantime, India's demand for HFCs is expected to rise dramatically (Purohit et al., 2016).

With a population exceeding one billion and a rapidly expanding economy, India's halocarbon emissions are expected to have global significance. Based on inferred consumption trends, Velders et al. (2015) estimated that India will emit 400 Tg CO$_2$eq yr$^{-1}$ of HFCs in 2050, a 67-fold increase over 2016 emissions. However, little else is known about India's emissions. Estimates from bottom-up, inventory-based methods have only been made for a subset of HFCs (HFC-134a, HFC-152a and HFC-23) in India and only up to 2010 (Garg et al., 2006; Ministry of Environment, Forest and Climate Change, 2012, 2015). With the exception of DCM, for which Leedham Elvidge et al. (2015) estimated India's emissions to be 20.3 (15.8 – 24.8) Gg

yr$^{-1}$ in 2008, emissions of these gases have never been estimated for India through regional 'top-down' or inverse modelling approaches that use atmospheric mole fraction measurements to infer surface fluxes. However, top-down methods have been applied elsewhere in Asia (Palmer et al., 2003; Yokouchi et al., 2005; Kim et al., 2010; Saikawa et al., 2012; Stohl et al., 2010; Lunt et al., 2018).

5     Previous studies in other countries have shown that there can be large discrepancies between national inventories of halocarbons and those inferred from atmospheric observations (Graziosi et al., 2017; Lunt et al., 2015; Say et al., 2016). Therefore, this dual quantification approach has been highlighted by many organizations as being beneficial for accurate and transparent greenhouse gas reporting (Leip et al., 2018). In this study, we present the first top-down estimates of India's halocarbon emissions and provide a 2016 benchmark, which is critical for evaluating future policy changes surrounding India's halocarbon 10  emissions.

## 2   Materials and Methods

### 2.1   Collection and analysis of air samples

Atmospheric samples were collected in evacuated 3 L stainless steel electro-polished flasks (SilcoCan, Restek, USA) aboard the UK's FAAM (Facility for Airborne Atmospheric Measurements) BAe-146 research aircraft. In total 176 samples were 15  collected over 11 flights conducted between the 12th June and 9th July 2016 (Table 2). On nine of these flights, samples were collected over northern India at altitudes ranging predominantly between 0 – 1.5 km (Fig. 1). Air was drawn through a forward-facing air sampling pipe on the exterior of the aircraft and pressurised into the sample flasks using a metal bellows pump (Senior Aerospace PWSC 28823-7). Sample flasks were evacuated to 1e$^{-5}$ psig prior to each flight. Before sample collection, the lines within each sample case were flushed with ambient air for a minimum of one minute. Sample flasks were filled to a maximum 20  pressure of 41 psig, giving a usable sample volume of 9 L at atmospheric pressure. Sample filling typically varied between 25 - 60 seconds in duration, depending on altitude (equivalent to ~7 km of flight track at average cruise velocity). Flasks were filled at regular intervals during each flight (interval dependent on flight length). When not in use, flask samples were stored in a container with no air-conditioning, to eliminate the risk of sample contamination from leaking air-conditioning refrigerant. None of the gases discussed here were present on the research aircraft itself, and the laboratory at the University of Bristol 25  does not contain a HFC filled air-conditioning unit. Apart from flasks collected over the Arabian Sea, samples were transported from India to Bristol within one month of collection.

    Flask samples were analysed using the Medusa GCMS analytical system, with modifications made to the analysis to account for the small volume and low pressure of the flask samples. In the set-up described previously (Miller et al., 2008; Arnold et al., 2012), atmospheric measurements were derived from 2 L samples, injected into the pre-concentration system at a flow rate of 30  100 cm$^3$ min$^{-1}$, resulting in a total injection time of 20 minutes. For this work, each measurement was derived from three 1.75 L analyses and injected into the analytical system at a flow rate of 50 cm$^3$ min$^{-1}$, resulting in a total injection time of 35 minutes. The analysis of each flask was bracketed by analyses of a quaternary reference gas, to account for short term drifts in detector sensitivity. Halocarbon mole fractions are reported relative to a set of gravimetrically prepared 'primary'

standards (Table 3), via a hierarchy of compressed real-air standards held in 34 L electro-polished stainless-steel canisters (Essex Industries, Missouri, USA). The working (quaternary) standard was compared to a tertiary tank on a roughly monthly basis. System blanks were conducted monthly, to quantify possible interferences from system leaks and carrier gas impurities. For each gas, the ratio of target to qualifier ion(s) was continually monitored to ensure that co-eluting species did not interfere with the analyses. For each flask, measurement precision was estimated as the standard deviation of the three replicate analyses. Average measurement precisions are shown in Table 3 and are comparable to the precisions reported previously by Miller et al. (2008).

## 2.2  Numerical Atmospheric Modelling Environment (NAME)

A Lagrangian particle dispersion model was used to quantify the influence of surface fluxes on each atmospheric measurement. The Met Office NAME (Numerical Atmospheric dispersion Modelling Environment) model was run in backwards mode (Manning et al., 2011) to generate 30-day air histories for every minute along each flight path (each minute represents approximately 7 km of the flight track at average speed). These air histories represent the sensitivity of a measurement to fluxes from the surface (defined as 0 - 40 meters above ground level). NAME was driven using meteorological output from the operational analysis of the UK Met Office Numerical Weather Prediction model, the Unified Model, with a horizontal resolution of approximately 17 km in 2016. The model domain spanned from $55 - 109$ °E and $6 - 48$ °N up to 19 kilometres altitude (Fig. S1). For each flight minute, tracer particles were released at a rate of 1000 particles $min^{-1}$ from a cuboid, whose dimensions were determined by the change in latitude, longitude and altitude of the aircraft during that one-minute period. In general, samples were collected during level sections of each flight path, minimising transport errors that could arise from releasing particles over a range of altitudes. At the boundaries of the domain, the three-dimensional location and time at which each particle left the domain were recorded to provide the sensitivity to boundary conditions.

Given the short lifetimes of DCM, PCE and chloroform there is some chemical loss during a typical 30-day simulation. Fang et al. (2018) investigated the impact of modelling short-lived substances with lifetimes of around six months over regional domains, without accounting for loss processes. Their study showed that, for sources that are within several hundred kilometres of measurement locations, as in this set-up, the decay is very small (less than 1%) over the time-scales of transport from source to receptor and can thus be neglected.

The ability of NAME to accurately simulate transport is critical for ensuring robust emission estimates. Model simulated wind direction and speed were compared to meteorological data recorded on board the FAAM aircraft (Fig. S2–S3). To ensure that transport errors had a minimal impact on the inversion, emissions derived using the complete set of atmospheric measurements were compared to those derived from a filtered dataset (Fig. S4), in which observations corresponding to periods where the NAME simulated wind speed/direction differed from the measured meteorology by more than 20% were removed.

## 2.3  Inverse modelling using atmospheric dispersion modelling

Our inverse method is based on the trans-dimensional approach described by Lunt et al. (2016). Emissions and uncertainties were characterized using principals of hierarchical Bayesian modelling detailed in Ganesan et al. (2014). The inverse approach

solves for a parameter vector, $\boldsymbol{x}$ (including flux fields and boundary conditions), using measurement data, $\boldsymbol{y}$. In a Bayesian framework, independent prior knowledge of emissions, $\boldsymbol{x}_{ap}$, is used in conjunction with measurements to solve for a posterior emissions distribution, $\boldsymbol{x}$ using a linear model, $\mathbf{H}$ (Eq. 1).

$$\boldsymbol{y} = \mathbf{H}\boldsymbol{x} + \epsilon \tag{1}$$

$\mathbf{H}$ is a Jacobian matrix of sensitivities, here describing the relationship between changes in atmospheric mole fractions and changes in the parameter vector $\boldsymbol{x}$. $\epsilon$ is uncertainty arising from the model and the measurements. In a traditional Bayesian inversion, uncertainty in $\boldsymbol{x}_{ap}$ and the model-measurement uncertainty, $\epsilon$, are both assigned prior to the inversion. These uncertainties are often poorly known and rely on a subjective decision by the investigator, but have been shown to significantly impact upon the derived posterior emissions (Peylin et al., 2002; Rayner et al., 1999). To minimize this impact, a hierarchical approach incorporates additional hyper-parameters, which allow for the propagation of 'uncertainties in these uncertainties' to the posterior solution.

$$\rho(\boldsymbol{x}, \theta | \boldsymbol{y}) \propto \rho(\boldsymbol{y} | \boldsymbol{x}, \theta) \cdot \rho(\boldsymbol{x} | \theta) \cdot \rho(\theta) \tag{2}$$

Eq. 2 is a hierarchical version of Bayes' theorem (normalizing factor not shown for brevity). In this example, the prior emissions uncertainty is governed by a hyper-parameter ($\theta$), which has a probability density function (PDF) that is explored within the inversion. This equation can also be employed in a similar way for the model-measurement uncertainty or any other unknown parameters. The hierarchical Bayesian approach was extended to a trans-dimensional framework, in which the number and configuration of the spatial grid over which emissions were estimated were also unknown parameters, prior to the inversion. Therefore, it is largely the information content of the measurements that govern these unknown aspects. This framework has been shown to result in a more robust and justifiable quantification of uncertainties in emissions than traditional approaches.

In general, Eq. 2 is not solvable via analytical means and was estimated using reversible jump Markov Chain Monte Carlo (rj-MCMC). The rj-MCMC algorithm was used to sample 320,000 variants of the parameter space with the first 120,000 discarded as 'burn-in' to ensure that the system had no knowledge of the initial state. The remaining 200,000 samples were then used to form the posterior PDFs. In our estimates, the means of these posterior PDFs are presented, with the uncertainties represented by the $5^{th}$ and $95^{th}$ percentile values.

Emissions were aggregated into totals for the northern-central India (NCI) region (Fig. 1), which contains 72% of India's population, and then extrapolated to a national total for all gases besides HFC-32, CTC, MCF and chloroform. The sources of the other gases except HFC-23 are refrigeration, foams, aerosols and landfills, for which we assume population to be a reasonable proxy for scaling emissions, however we are not able to quantify the uncertainty associated with extrapolating to a national total without additional measurements. For HFC-23, the NCI region incorporated four of the five known manufacturing plants for HCFC-22. To estimate national emissions, we scaled the NCI total by the ratio of HCFC-22 produced at those four

factories, to total production at all five (based on 2015 factory specific production statistics (UNEP, 2017)). Based on these statistics, over 98% of HCFC-22 was produced by factories residing within the NCI.

While the estimates presented here represent emissions over a two-month period, they are likely to be consistent with annual emissions for gases that are not expected to have significant seasonality in India. Seasonal variations in emissions have been observed in HCFC-22 and HFC-134a in Western Europe and North America (Xiang et al., 2014), showing that summertime emissions are two and three times larger than wintertime emissions for the two gases, respectively. The authors attribute this seasonality to increased vapour pressure in sealed refrigeration/air-conditioning systems as a result of higher ambient temperatures, and to increased use of such systems during summer months. While some degree of seasonality might be expected for India's emissions of these gases, is not possible to estimate the magnitude of seasonality without long-term observations from the Indian sub-continent. Our estimates for HCFC-22 and HFC-134a should be considered representative of June-July 2016 until long-term studies are conducted. Biogenic sources of chloroform have also been shown to exhibit seasonality (Laturnus et al., 2002), yet emissions from anthropogenic activities (e.g. use as a feedstock) are not likely to vary by season. No such seasonality has been reported for any of the other gases discussed here.

Due to sampling by aircraft, our estimates are likely to be representative on a regional-scale for gases that have sources that are widespread and do not vary significantly in time throughout the measurement period. These characteristics are thought to be true for most gases studied here. With the exception of HFC-32, HFC-23, CTC, MCF and chloroform, emissions of the other gases are expected to be dominated by sources linked to consumption (Wan et al., 2009; McCulloch et al., 2003), as opposed to production. Production could have short-term variations in emissions rate due to, for example, facility down-time. We also discuss below that some caution must be made in the interpretation of HFC-125 emissions.

## 2.4   A priori emissions

A priori emissions were assembled from a variety of sources owing to the limited information available for India. **CFCs**: Since a total ban on CFC production and consumption has been in place since 2010, country specific emissions/consumption data no longer exist. Despite this, studies suggest that emissions of these gases could be ongoing (Montzka et al., 2018). To estimate a priori total emissions over India, we scaled an estimate of 2016 global emissions derived using the AGAGE 12-box model (an extension of Rigby et al. (2014), see section 2.6) by population (though CFC emissions are not necessarily expected to distribute globally according to population due to differences in Article 5 versus non-Article 5 country emission trends, amongst other factors). **HCFCs**: A priori total HCFC emissions over India were based on 2015 consumption data reported by India in its HPMP Stage II Road Map report (Ministry of Environment, Forest and Climate Change, 2017). Consumption is likely an underestimate of emissions due to the presence of banked sources such as refrigerators and foams. **HFCs**: Excluding HFC-23, prior HFC emission totals for India were calculated by scaling the 2010 EDGAR v4.2 (European Commission, 2009) Asian continental total by population (with India accounting for approximately 29% of the Asian total). This was done because EDGAR does not indicate any emissions from India. For HFC-23, prior emission totals for India were based on the 2010 HFC-23 emissions reported in India's Biennial Update Report to the United Nations Framework Convention on Climate Change (Ministry of Environment, Forest and Climate Change, 2015), and extrapolated to 2016 using reported HCFC-22 production

data (and assuming a constant co-production ratio) (Ministry of Environment, Forest and Climate Change, 2017). **Regulated chlorocarbons**: India's CTC emissions were estimated at 2.8 Gg yr$^{-1}$ based on the 2014 estimate by Sherry et al. (2018). As with the CFCs, a priori MCF emissions were calculated using a population-based scaling of the global total derived using the AGAGE 12-box model, and hence estimated to be 0.3 Gg yr$^{-1}$ in 2016. **Unregulated chlorocarbons**: For DCM, a priori

emissions were from Leedham Elvidge et al. (2015), which estimated India's DCM emissions to be 20.3 Gg yr$^{-1}$ in 2012 based on independent measurements. India's PCE emissions were from the Reactive Chlorine Emissions Inventory (McCulloch et al., 1999) and were estimated at 6.0 Gg yr$^{-1}$. Terrestrial chloroform emissions were taken from the AGAGE 12-box model. Using a population scaling for India and assuming that 45% of chloroform emissions (biogenic and anthropogenic) originate on land (McCulloch, 2003), India's land-based chloroform emissions were estimated at 3.0 Gg yr$^{-1}$. Oceanic chloroform emissions

were adapted from Khalil et al. (1999), who estimated a northern hemispheric tropical ocean source of 50 Gg yr$^{-1}$. The ocean within our model domain was estimated to account for 8.3% of this source by area, equivalent to 4.2 Gg yr$^{-1}$.

No recent spatial information was available for any of the halocarbons studied here. With the exception of chloroform, for which prior emissions were distributed uniformly across both land and ocean, prior emissions totals were distributed across the model domain using the National Oceanic and Atmospheric Administration (NOAA) DMSP-OLR satellite night light data,

available at 30 arc second resolution (https://ngdc.noaa.gov/eog/data/web_data/v4composites/). The night lights distribution was a useful starting point for our emissions maps, since night lights are generally correlated with population density (Raupach et al., 2010), but are also likely to include industrial sites, such as HCFC-22/chloromethane manufacturing plants. We expected the major sources of CFCs, HCFCs, HFCs and chlorocarbons to be explicitly linked to domestic and/or commercial activities, or from industries requiring a significant work-force.

For all species, the prior emissions uncertainty was described by a uniform PDF with lower and upper bounds of 50% and 500% respectively. This large uncertainty reflects the lack of detailed information available for India. Large prior uncertainties mean that our posterior emissions over the NCI are informed almost entirely by the atmospheric measurements. To confirm that our posterior estimates were independent of the spatial distribution of the prior, results derived using the night-lights data were compared to those derived from a spatially uniform prior (Fig. S4).

**2.5   A priori boundary conditions**

The footprints from NAME only model the emissions released within the model domain. Hence, a prior estimate of the mole fraction at the boundaries of model domain must be made and incorporated into the modelled mole fraction. Mole fraction 'curtains' of each gas were used to provide a priori information about boundary conditions (it should be noted that these boundary conditions were adjusted within the inversion). For the HFCs, mole fractions were simulated using the 3D global

chemical transport model MOZART (Model for OZone and Related chemical Tracers (Emmons et al., 2010)). MOZART was driven by offline meteorological fields from MERRA (Modern Era Retrospective-Analysis for Research and Applications (Rienecker et al., 2011)). For the CFCs, HCFCs and chlorocarbons, MOZART fields were not available, and uniform curtains were assumed. The mole fraction for each curtain in each month was estimated using the AGAGE 12-box model (Rigby et al., 2014) and measurements from five baseline AGAGE observatories. For each gas, the model was used to estimate a monthly

baseline mole fraction for four latitude bands. The simulated mole fraction from latitude bands 30 – 90 °N, 0 – 30 °N and 0 – 30 °S were used to assign a priori mole fractions to the northern, eastern/western and southern curtains of the model domain respectively. The boundary conditions associated with each NAME-simulated measurement were calculated by mapping the exit times and locations of particles leaving the domain to the curtains. In addition to emissions parameters, a decomposition of the a priori boundary conditions, represented as offsets to the curtains in the four directions, were also solved for in the inversion.

## 2.6 Global halocarbon emissions estimation

Indian halocarbon emissions were compared to global emission estimates calculated using the AGAGE 12-box model (Rigby et al., 2014), assimilating data from five remote AGAGE background sites (Mace Head, Ireland; Trinidad Head, USA; Ragged Point, Barbados; Cape Matatula, American Samoa and Cape Grim, Tasmania) following a Bayesian inversion methodology. Baseline monthly means were estimated by statistically filtering the high-frequency data (O'Doherty et al., 2001). The data were averaged into semi-hemispheres (30 °N – 90 °N, 0 °N – 30 °N, 30 °S – 0 °S, 90 °S – 30 °S) for comparison with mole fractions predicted by the AGAGE 12-box model, which resolves these four semi-hemispheres, with vertical levels separated at 500 and 200 hPa (Cunnold et al., 1983; Rigby et al., 2013). The model uses annually repeating meteorology and OH concentrations from Spivakovsky et al. (2000), tuned to match the growth rate of methyl chloroform.

Total atmospheric lifetimes (Table 1) were estimated using the halocarbon-hydroxyl temperature-dependent rate constants from Burkholder et al. (2015) (tropospheric removal) and the average photochemical model loss frequencies given in Ko et al. (2013) (stratospheric removal). A Bayesian framework was used to derive emissions from the data and the model, in which an a priori estimate of the emissions growth rate was adjusted to bring the model into agreement with the data (following Rigby et al. (2011)). The inversion propagates uncertainties in the observations through to the derived fluxes and augments the derived fluxes with uncertainties due to the lifetime and potential errors in the calibration scale. These estimates are a 2016 extension of those presented in Rigby et al. (2014).

## 3 Results

### 3.1 Atmospheric Measurements

Measurements were made from whole air flask samples collected over India during June and July 2016. Fig. 1 shows the location and altitude of these measurements along with the model-derived sensitivity of these samples to surface emissions. Mole fractions of each halocarbon measured during the campaign are shown in Fig. 2. For comparison, each gas is shown alongside baselines representative of the Northern and Southern Hemispheres. These baselines were derived from statistical fits to observations from the AGAGE sites at Mace Head, Ireland and Cape Grim, Tasmania (Prinn et al., 2018). Although all of the samples collected during our campaign were within the Northern Hemisphere, the South Asian monsoon, which occurs annually between June and September, draws air from southern latitudes, resulting in the Indian regional background

being more consistent with the Southern Hemisphere at this time of year. For CFC-12 and CFC-113, owing to the decrease in emissions resulting from the Montreal Protocol, the hemispheric baselines are now similar and the difference in mole fraction between hemispheres is smaller than the average precision of our flask measurements.

Enhancements in mole fractions over the regional background form the basis for estimating regional emissions. For all species except HFC-134a, the average mole fractions of samples collected over the Arabian Sea were lower than those collected directly over NCI. Variability in the mole fraction of samples collected over NCI varied considerably by species. For CFC-11, CFC-12 and CFC-113, few pollution events were observed, and their signals were of similar size to the measurement precision. Similarly, only small enhancements were observed for HCFC-142b, suggesting its main use as a foam-blowing agent was not significant or was not widespread and thus could not be discerned in the aircraft samples.

In contrast, large enhancements in mole fraction were observed for HFC-134a and HCFC-22, suggesting that usage of these substances as a refrigerant is widespread. It is likely that these gases share a range of common sources, including use in India's largest refrigeration and air-conditioning sector, stationary air-conditioning (Purohit et al., 2016), though the rate of transition from HCFC to HFC could vary by region. We find a significant (R = 0.53, Fig. 3) relationship between HFC-134a and HCFC-22 mole fractions, consistent with some co-located sources. Large enhancements in HFC-23 mole fraction suggest that the samples were sensitive to emissions from HCFC-22 manufacturing facilities, as HFC-23 is a by-product of HCFC-22 production. The NCI region contains four out of the five Indian manufacturing facilities that were registered under the Clean Development Mechanism (CDM) (https://cdm.unfccc.int/Projects/registered.html).

Enhancements were also observed in HFC-32 and HFC-125, although the observed mole fractions for these species were not strongly correlated (R = 0.15, Fig. 3), suggesting that India is yet to adopt refrigerant blend R-410A (50% by wt. HFC-125, 50% by wt. HFC-32) on a large scale. Conversely, atmospheric measurements from China are consistent with widespread use of R-410A after 2010 (Li et al., 2011; Yao et al., 2012; Wu et al., 2018), suggesting that India lags behind China in the uptake of the HFC blends designed to replace HCFC-22, or that it has adopted lower GWP alternatives. Similar to China, HFC-125/HFC-32 measurements at Mace Head (Ireland) over the same time-period were strongly correlated (R = 0.86). All enhancements in HFC-32 are found to correspond with enhancements in DCM (Fig. 3), suggesting that India's emissions of this gas are linked to its production. The significance of this correlation is discussed further in section 3.2.3.

We found no correlation for HFC-125 and HFC-143a (R = -0.04, Fig. 3), gases whose emissions are regularly linked through the consumption of blends R-404A (52% by wt. HFC-143a, 44% by wt. HFC-125, 4% by wt. HFC-134a) and R-507A (50% by wt. HFC-125, 50% by wt. HFC-143a) (Montzka et al., 2014; O'Doherty et al., 2014). In contrast, at Mace Head, a strong HFC-125/HFC-143a correlation (R = 0.78) was observed during this time.

Evidence for widespread use of both HCFCs and HFCs and the lack of large enhancements in CFCs suggests that India's transition to first- and second-generation CFC replacements is nearing completion. However, there appears to be little evidence for the consumption of HFC blends or HFC-152a in 2016, refrigerants/propellant used extensively in the developed world (Greally et al., 2007; O'Doherty et al., 2014).

Only a small number of enhancements were observed for the regulated chlorocarbons CTC and MCF, while a large number of enhancements were observed for all three unregulated chlorocarbons. In particular, very large enhancements were found

for DCM, with a maximum mole fraction of 1133 ppt (corresponding enhancement of 1120 ppt, Fig. 2). Samples collected at longitudes east of 81 °E were particularly enhanced above the baseline, suggesting that the flask samples were sensitive to regions producing/consuming large quantities of DCM as a solvent, feedstock or both.

We found a significant (R = 0.71) correlation between DCM and chloroform (Fig. 3), suggesting that these gases share some similar sources or source locations (i.e. DCM and chloroform are chloromethanes manufactured for use as feedstock gases for HFC-32 and HCFC-22, respectively). Since DCM is predominantly anthropogenic in origin, this correlation indicates that some of the enhancements observed for chloroform are from anthropogenic sources. However, we find a low correlation (R = 0.24) between HFC-23 and chloroform (Fig 3). This suggests that either fugitive losses during chloroform manufacture are not co-incident with losses during HCFC-22 production, or there are also other sources of chloroform such as biogenic sources.

During two flights (B959 and B963) conducted on the $21^{st}$ and $25^{th}$ of June, a small number of samples were collected over the Arabian Sea. NAME back-trajectory analysis was used to show that these samples had not interacted with any significant landmass in the 30 days prior to collection. Despite this, four of the six samples collected on these flights exhibited an elevated HFC-134a concentration, which did not correlate with any other species, including HCFC-22. One possible explanation for enhancements only being observed in HFC-134a over the Arabian Sea is that they are the result of sporadic emissions from ship-based air-conditioning systems, since all Arabian Sea samples were collected at low altitude (0.01 - 0.8 km).

### 3.2 Halocarbon emissions estimates for NCI and India

Mean NCI and Indian emissions estimates and the relative contributions of each gas to 2016 global emissions are shown in Fig. 4 and tabulated in Table 4 (Gg yr$^{-1}$) and Table 5 (Tg CO$_2$eq yr$^{-1}$). Uncertainties presented throughout correspond to the $5^{th}$ and $95^{th}$ percentiles of the posterior distribution.

We estimate India's 2016 CFC, HCFC and HFC (excluding HFC-32) emissions to be 54 (27 – 86) Tg CO$_2$eq yr$^{-1}$, 15 (11 - 19) Tg CO$_2$eq yr$^{-1}$ and 53 (40 – 67) Tg CO$_2$eq yr$^{-1}$ respectively, which correspond to 7 (4 – 12) %, 2 (1 – 3) % and 6 (5 – 8) % of global emissions. Combined emissions of regulated (CTC and MCF) and unregulated (DCM, PCE and chloroform) chlorocarbons from NCI are estimated at 11 (7 - 16) Tg CO$_2$eq yr$^{-1}$ and 1 (1 - 2) Tg CO$_2$eq yr$^{-1}$, which account for 7 (4 - 10) % and 8 (6 - 9) % of global emissions, respectively. With the exception of DCM, there are no previous top-down national-scale estimates of any of these gases for India. In 2016, India's aggregated HFC emissions were approximately an order of magnitude larger than the 2016 emissions assumed by Velders et al. (2015), suggesting that future projections of India's HFC emissions could be inaccurate.

### 3.2.1 CFCs

Through commitments under the Montreal Protocol, India finalised its phase-out of the consumption and production of CFCs in 2010. However, residual emissions from banks (refrigerators, foams, landfills etc.) are expected to continue for several decades (Rigby et al., 2014). Our mean CFC-11, CFC-12 and CFC-113 emissions are 1.7 (0.8 – 3.1) Gg yr$^{-1}$, 4.1 (2.1 – 6.3) Gg yr$^{-1}$ and 0.5 (0.2 – 0.8) Gg yr$^{-1}$, respectively, corresponding to 2 (1 - 4) %, 13 (7 - 20) %, and 7 (2 - 11) % of global emissions in 2016. The magnitude of the uncertainties in our CFC estimates are largely a reflection of the precision of the measurements.

Further work is needed through additional high-precision measurements, particularly for CFC-12, to narrow this uncertainty. For CFC-11, our 2016 estimate of 1.2 (0.6 – 2.2) Gg yr$^{-1}$ suggests that NCI, the region with the majority of India's population, is unlikely to have contributed significantly to the recent rise in global emissions (an increase of 13 $\pm$ 5 Gg yr$^{-1}$) reported between 2013 – 2016 (Montzka et al., 2018).

### 3.2.2 HCFCs

There is limited information about HCFC emissions from India, with the current state of knowledge encapsulated only in reports of production and consumption (Ministry of Environment, Forest and Climate Change, 2017). We find that India's 2016 HCFC emissions are dominated by HCFC-22 at 7.8 (6.0 – 9.9) Gg yr$^{-1}$, and these emissions comprise only 2 (1 - 3) % of global emissions. Estimating seasonal variations in emission rate for India is not possible without long-term observations. Hence, our estimate for this gas should be considered representative of the measurement period only. Our HCFC-22 emissions are comparable in magnitude to HFC-134a and HFC-125, discussed below, suggesting that India's transition from HCFCs to their non-ozone depleting replacements is in progress. India's HCFC-22 emissions are considerably smaller than those from other nations such as China, whose emissions in 2007 were estimated at 165 (140 - 213) Gg yr$^{-1}$ (Vollmer et al., 2009) and the USA, whose emissions in 2014 were estimated at 40.0 (34.1 – 45.8) Gg yr$^{-1}$ (Hu et al., 2017).

Our estimates of India's HCFC-141b and HCFC-142b emissions are small (1.0 (0.7 – 1.5) Gg yr$^{-1}$ and 0.10 (0.06 – 0.14) Gg yr$^{-1}$, respectively). Taken together with the small reported consumption of these gases in 2015, our results suggest that either these substances have not had widespread usage in India or that efforts have been made by India under Stage I of the HPMP to phase-out HCFC consumption in the foam sector (Ministry of Environment, Forest and Climate Change, 2017), in favour of zero-ODP alternatives (UNDP, 2013). However, without detailed emissions information from previous years, it is not possible to determine whether the latter has been in effect.

### 3.2.3 HFCs

India's HFC emissions are dominated by emissions of HFC-134a and HFC-125, with estimated rates of 8.2 (6.1 – 10.7) Gg yr$^{-1}$ and 6.4 (5.2 – 7.8) Gg yr$^{-1}$ respectively. These emissions correspond to 4 (3 - 5) % and 10 (8 - 12) % of global emissions. Previous studies reported seasonality in emissions of HFC-134a from Western Europe and North America. Without long-term measurements to quantify this seasonality in India, our emissions rate should only be considered representative of the measurement period.

There are significant discrepancies between previous bottom-up estimates and our top-down results. Garg et al. (2006) estimated Indian HFC-134a emissions to be 1.1 Gg yr$^{-1}$ in 2005, while India reported 0 Gg yr$^{-1}$ in 2010 in its Biennial Update Report to the United Nations Framework Convention on Climate Change (UNFCCC) (Ministry of Environment, Forest and Climate Change, 2015).While there are no top-down comparisons for 2005, our results show there could have been significant growth in emissions of HFC-134a since 2005 and/or large discrepancies between bottom-up and top-down methodologies.

Further work and additional measurements are required to better understand the non-refrigerant blend sources of HFC-125. Our results suggest a possible application of HFC-125 in India as a standalone refrigerant, or an application that is not

currently for HCFC-22 replacement. Possible contributors are fire suppression, use as a solvent and the production of HFC-125 for export. While our model was able to capture most of the signals for the gases studied here (Fig. 5), it was unable to simulate some of the elevated measurements for HFC-125, indicating that in addition to widespread, constant sources, there could be point sources of HFC-125 that are episodic and difficult to resolve in a model.

5    We estimate India's emissions of HFC-143a to be 0.8 (0.4 – 1.2) Gg yr$^{-1}$, which comprise 3 (1 - 4) % of global emissions. Our low HFC-143a estimate corroborates our assertion of minimal R-404A and R-507A consumption. There are no previous estimates for Indian HFC-143a emissions.

India's HFC-152a emissions are estimated to be 1.2 (0.9 – 1.4) Gg yr$^{-1}$, which amount to 2 (2 - 3) % of global emissions. Garg et al. (2006) estimated India's HFC-152a emissions to be 0.04 Gg yr$^{-1}$ in 2005 and attributed these to the glass industry. 10  Our emission rate is comparatively large, suggesting that either there are discrepancies with inventory methodologies or that there has been substantial growth in emissions in the last decade. Regardless, these emissions are small compared to other countries, particularly China, whose emissions of HFC-152a were estimated at 16 Gg yr$^{-1}$ in 2013 (Fang et al., 2016), and the USA, for which an emission rate of 51.5 (35.5 – 75.5) Gg yr$^{-1}$ was estimated for 2012 (Simmonds et al., 2015).

HFC-32 emissions are estimated for NCI to be 0.44 (0.36 – 0.54) Gg yr$^{-1}$. All the measured enhancements in HFC-32 15  are correlated with enhancements in DCM, a feedstock in the manufacture of HFC-32 (Fig. 3). These measurements suggest that India's HFC-32 emissions originate predominantly from fugitive losses during the manufacturing process, rather than widespread use in a refrigerant blend. Our assertion is consistent with a previous study (Leedham Elvidge et al., 2015), which attributed growth in South Asian emissions of DCM to HFC-32 manufacture. Since our NCI HFC-32 estimate is attributed to production, we consider it to be decoupled from population density, and hence we have not scaled this value to a national total. 20  In addition, given emissions from the manufacturing process could vary in time (e.g. as a result of facility down-time), our emissions estimate for this gas should be considered representative of the measurement period only.

### 3.3   India's HFC-23 emissions and the Clean Development Mechanism

HFC-23 emissions are estimated for India to be 1.2 (0.9 – 1.5) Gg yr$^{-1}$, which comprise 10 (7 - 12) % of global emissions. Emissions of HFC-23 are linked to production of HCFC-22 and could vary in time due to unforeseen facility downtime or fluc- 25  tuations in demand for HCFC-22. However, based on data reported under the CDM (https://cdm.unfccc.int/Projects/registered.html), there is evidence to suggest that HCFC-22 production rates have in previous years remained relatively constant over any given year. While we therefore assume that our estimate is representative of an annual average, further measurements are required to fully evaluate any short-term variability in emissions of HFC-23. Fig. 6 shows that emission 'hot-spots' picked out by the inverse model are consistent with the known locations of HCFC-22 manufacturing facilities.

30  Between 2004 and 2013, India received substantial funding from the Clean Development Mechanism for the abatement of HFC-23 produced during the manufacture of HCFC-22. To assess the impact of the CDM on India's HFC-23 emissions, we compare our HFC-23 emission estimate to previous estimates derived from bottom-up methods (Fig. 7). Emissions between 1990 and 2005 are from Garg et al. (2006), and in 2007 and 2010 are from India's reports to the UNFCCC (Ministry of Environment, Forest and Climate Change, 2012, 2015). The reported bottom-up estimates show accelerating growth in India's

HFC-23 emissions, which increased from 0.07 Gg yr$^{-1}$ in 1990 to 1.43 Gg yr$^{-1}$ in 2010. It is important to note that there is a large discrepancy in emissions reported in the UNFCCC inventory and in the manufacturers' CDM submissions, suggesting inconsistencies in the two methodologies. These discrepancies highlight the value of independent top-down estimates.

Depending on the efficiency of the manufacturing process, the HFC-23/HCFC-22 production ratio can vary between 0.014 (Rotherham, 2004) for optimised processes to values in excess of 0.04 for inefficient processes (McCulloch and Lindley, 2007). The production ratio is equal to the quantity of HFC-23 produced with respect to the quantity of HCFC-22 produced and is equivalent to an HFC-23 emission ratio when no abatement technologies are implemented. Based on India's HCFC-22 production statistics and bottom-up HFC-23 emission estimates, in 2007, prior to when all five manufacturers of HCFC-22 reported the use of abatement technologies, the average production ratio was 0.031.

The Clean Development Mechanism was in operation in India between 2004 and 2013. During the period of the CDM when abatement was in use at all facilities (2009 - 2013), the average emission ratio dropped to 0 – 0.009 based on the amount of non-abated HFC-23 (i.e. emissions vented to the atmosphere) reported by the manufacturing facilities. Our top-down estimate in 2016 corresponds to an average emission ratio of 0.022. While the CDM may have been effective in reducing HFC-23 emissions, our results are consistent with resumed venting of HFC-23 by some or all manufacturers, following the discontinuation of CDM funding. In October 2016, the Indian government issued a national order requiring all manufacturers of HCFC-22 to maintain a proven abatement system and ensure capacity for the storage of HFC-23 for abatement system down-time. With such systems in place, possible growth in India's HCFC-22 production rate might not result in increased emissions of HFC-23.

### 3.4 Regulated chlorocarbons

We estimate CTC emissions from NCI to be 2.3 (1.5 - 3.4) Gg yr$^{-1}$, which accounts for 7 (4 - 10) % of global emissions in 2016. India reported that its production and consumption of CTC had ceased prior to 2016 (http://ozone.unep.org/countries/data). Since then, ongoing CTC emissions from India may not be linked to its use as solvent but may persist due to fugitive leaks during chloromethane manufacture (most notably DCM and chloroform production) and from chlorine consuming industries, such as chlor-alkali plants. India does not have any major operational facilities for the manufacture of PCE, which is another anthropogenic source of CTC (Sherry et al., 2018). As these activities are not thought to be distributed evenly with respect to population, we do not scale NCI estimates to a national total.

Sherry et al. (2018) estimated that India's chloromethane manufacturers might have produced as much as 20 Gg of CTC as by-product in 2014, with corresponding fugitive emissions of 1.8 Gg yr$^{-1}$, though these findings are not reflected in the UNEP reports. CTC is typically produced as a by-product of chloromethane (DCM and chloroform) manufacture at an estimated rate of 4%, and the ratio of DCM to chloroform production, while variable, typically varies from 30:70 to 70:30 (Oram et al., 2017; Sherry et al., 2018). Hence, if the total production of DCM/chloroform is known, the quantity of CTC produced may also be inferred. While we were unable to find any chloromethane production data for India, chloroform is used as a feedstock in the production of the refrigerant HCFC-22. Over 99% of chloroform produced globally is used in the manufacture of HCFC-22, with 1 kg of HCFC-22 requiring 1.5 kg of chloroform as feedstock (Oram et al., 2017). Based on an extrapolation of reported

HCFC-22 production statistics in India (available from 2006 – 2015 (UNEP, 2017)), we estimate India's HCFC-22 production in 2016 to be 55 Gg. If all chloroform produced was used for HCFC-22 manufacture, and all demand was met domestically (available data suggests India only imported ~165 tonnes of chloroform in 2016 (https://www.seair.co.in/chloroform-import-data.aspx)), we estimate that India would produce 82.5 Gg of chloroform in 2016. Based on the possible DCM/chloroform
production ratios discussed above, India is estimated to have produced 117 - 275 Gg of chloromethanes, and hence 4.7 - 11.0 Gg of CTC, in 2016. Since the majority of India's chloromethane manufacture occurs within NCI, this suggests that a significant amount of CTC is either destroyed or sold for non-dispersive applications. One such application is the production of divinyl acid chloride (DVAC). Sherry et al. (2018) estimated that India's DVAC industry consumed 20 Gg of CTC in 2014.

Our posterior emissions map (Fig. 6) shows that the majority of CTC emissions originate from chloromethane manufacturing
facilities, while the known locations of chlor-alkali plants do not appear to be associated with large emissions. The emissions distribution of CTC resulting from the inversion is similar to that of HFC-23. This may be because CTC is a by-product of chloromethane (i.e., chloroform and DCM) manufacture, and HFC-23 is produced during the manufacture of HCFC-22, which requires chloroform as a feedstock. The locations of the main HCFC-22 production facilities are in similar locations to the chloromethane facilities in NCI (Fig 6). India's CTC emissions remain small compared to those of eastern China, whose
average emissions from 2011 - 2015 were estimated at 17 (11 – 24) Gg yr$^{-1}$ (Lunt et al., 2018), but are of similar magnitude to those of the US, estimated at 4.0 (2.0 – 6.5) Gg yr$^{-1}$ between 2008 – 2012 (Hu et al., 2016). Ongoing US emissions were attributed to industrial sources, particularly chlor-alkali plants, which differs from our finding that CTC emissions in India do not correspond with known locations of chlor-alkali production.

Based on its reports to the UNEP, India has not produced or consumed MCF since 2001. However, a small number of
enhancements in the mole fraction of this gas suggests that sources persist. Reimann et al. (2005) proposed that factories producing HCFC-141b and HCFC-142b were possible sources of MCF in Europe, since MCF is used as a feedstock in the production of these refrigerants. However, India does not report production of either HCFC-141b or HCFC-142b (Ministry of Environment, Forest and Climate Change, 2017). Landfills are another possible source of MCF, with previous studies from other regions reporting emissions from municipal waste disposal facilities (Maione et al., 2014; Talaiekhozani et al., 2018).
Therefore, the nature, location and magnitude of the sources of MCF are uncertain and we do not estimate a total for the whole of India. At 0.07 (0.04 - 0.10) Gg yr$^{-1}$, MCF emissions from the NCI account for 4.1 (2.4 – 5.9) % of global emissions. Despite its status as a developing country, which meant India had more time to phase-out consumption of MCF compared to developed countries, emissions from the NCI (which comprises 72% of India's population and includes several key industrial regions) are smaller than those from Europe, which were estimated to be 0.20 Gg yr$^{-1}$ in 2012 (Maione et al., 2014). Given the
continued role of MCF in estimating global hydroxyl concentrations (e.g. Rigby et al. (2017)), further long-term measurements from India are required to better understand the remaining sources of this gas.

### 3.5   Unregulated chlorocarbons

We estimate Indian DCM emissions to be 96.5 (77.8 - 115.6) Gg yr$^{-1}$, and these contribute 11 (9 - 13) % of global emissions. India's DCM emissions are small compared to the 455 $\pm$ 45.5 Gg yr$^{-1}$ emitted from China in 2015 Oram et al. (2017). When

compared to previous estimates of India's DCM emissions, our results reflect substantial growth. Leedham Elvidge et al. (2015) estimated emissions of 4.9 (2.7 - 7.2) Gg yr$^{-1}$ in 1998, rising to 20.3 (15.8 - 24.8) Gg yr$^{-1}$ in 2008, suggesting a 2- to 4-fold increase in emissions over that period. Our mean estimate represents an approximate 5-fold increase in emissions between 2008 and 2016. Global emissions over the same period rose from 611.5 Gg yr$^{-1}$ to 907.3 Gg yr$^{-1}$, representing an increase of

295.8 Gg yr$^{-1}$. The growth in India's emissions over this period (48.9 Gg yr$^{-1}$) would therefore represent 25.8% of the global rise. The rise in India's DCM emissions could possibly be attributed to increased production of HFC-32, however, no HFC-32 production information from India is available. Our HFC-32 measurements suggest that a large proportion of the HFC-32 produced by India is exported rather than consumed.

Emissions of PCE are almost exclusively anthropogenic in origin, due to its widespread use as a chemical intermediate

and general-purpose solvent. Despite classification as a hazardous air pollutant by the United States Environmental Protection Agency (EPA, 2012), PCE is used extensively in India as a dry-cleaning solvent (Srivastava, 2010). We estimate India's PCE emissions to be 2.9 (2.5 - 3.3) Gg yr$^{-1}$, which account for 4 (3 - 4) % of the global total. When compared to the only previous estimate of India's PCE emissions, which was calculated using bottom-up methods (3.9 Gg yr$^{-1}$ in 1990 (McCulloch et al., 1999)), our estimate either shows a discrepancy with bottom-up inventories or a decrease in emissions since 1990. The latter

would be consistent with global emissions derived using the AGAGE 12-box model (Rigby et al., 2014), which also show a decline from 124.2 (50.3 – 204.2) Gg yr$^{-1}$ in 2006 to 82.6 (35.8 – 133.3) Gg yr$^{-1}$ in 2016.

Because India's chloroform emissions are linked to industrial processes (chloromethane and HCFC-22 manufacture) and biogenic emissions, we do not scale to a national total. We estimate NCI chloroform emissions to be 32.2 (28.3 – 37.1) Gg yr$^{-1}$, and these emissions account for 10 (8 – 11) % of global emissions in 2016. However, given the large biogenic component

of global emissions, the contribution of the NCI to global anthropogenic emissions may be significantly larger.

## 3.6  Sensitivity tests

We performed two sensitivity tests. We assessed the sensitivity of derived emissions to the a priori emissions field. We also assessed the effect of inaccurate transport modelling on derived emissions by using a second, filtered dataset, removing times where NAME wind direction and wind speed differed by more than 20% from the measured parameters. A comparison of the

three posterior estimates is given in Fig. S4. For all 17 halocarbons, the three estimates are statistically consistent, indicating that our estimates were robust to the prior spatial distribution and to any small model transport errors.

## 4  Conclusions

We present the first national-scale top-down emissions estimates of halocarbons for India. We show that India's 2016 halocarbon emissions reflect low emissions of CFCs and regulated chlorocarbons CTC and MCF, and large emissions of HCFCs,

HFCs and unregulated chlorocarbons such as DCM. India reported a complete phase-out of its production of CFCs, CTC and MCF by 2010, however banks such as dated refrigeration equipment and insulating foams, as well as fugitive emissions from industry, may persist. Our results indicate that India's remaining major CFC emissions represent 7 (4-12) % of global

emissions. Of the refrigerant gases, India's largest emissions are from HFC-134a, HFC-125 and HCFC-22. HFC-134a and HCFC-22 have similar magnitudes of emissions, suggesting that India is in transition between employing HCFC and HFC refrigerants. We present evidence to suggest that India is yet to adopt several common refrigerant blends, including R-410A, R-404A and R-507A, all of which are used extensively in the developed world. India's apparent lack of uptake of refrigerant blends presents an opportunity for future climate mitigation strategies; if India can be encouraged to bypass HFCs in favour of low-GWP alternatives, substantial $CO_2eq$ emissions could be avoided. We also show that following discontinuation of funding from the CDM, some or all of India's manufacturers of HCFC-22 likely resumed venting of the HFC-23 by-product.

Our results indicate that small sources of MCF remain in India and we present evidence that India's CTC emissions are likely a by-product of chloromethane (DCM and chloroform) manufacture. Interest in the global emissions of unregulated chlorocarbons such as DCM, chloroform and PCE has grown in recent years, as increasing emissions from Asia pose a potential threat to the recovery of the ozone layer. Our DCM emissions estimate suggests a 5-fold increase in India's emissions since 2008.

As India's economy expands, its production and consumption of halocarbons is likely to increase dramatically. It is important to implement long-term and continuous halocarbon monitoring from this region of the world to help India evaluate its progress under the Montreal Protocol. Our 2016 estimates provide a benchmark, against which future changes to India's halocarbon emissions can be assessed.

## 5 Data availability

Data are available from the Centre for Environmental Data Analysis (CEDA):
http://catalogue.ceda.ac.uk/uuid/e838a628dacc438ab4749b011ae7225f

## 6 Code availability

Hierarchical Bayesian trans-dimensional MCMC code is available on request from Anita Ganesan (Anita.Ganesan@bristol.ac.uk).

*Author contributions.* D.S. collected the samples, conducted the instrumental analyses, ran the inverse model and wrote the paper. A.G. coordinated the flight campaign, co-developed the inverse model code and aided in the writing of the paper. M.L. co-developed the inverse modelling code and conducted the MOZART model runs. M.R. aided the inverse modelling work. S. O'D. co-conducted the instrumental analyses and provided data from Mace Head. C.H. produced the SIO calibration scales for these gases. A.M. advised on NAME modelling. P.K. provided measurements from Cape Grim. S.B. co-collected the whole air samples.

*Competing interests.* The authors declare that they have no conflict of interest.

*Acknowledgements.* We thank FAAM, Directflight and Avalon Aero personnel for their support during the flight campaign, and the site operators at the Mace Head and Cape Grim stations. We also acknowledge the contribution of the Ministry of Earth Sciences, Government of India, the UK National Environmental Research Council (NERC) and the Monsoon program's UK and Indian principal investigators. Daniel Say was funded by a NERC studentship. Anita Ganesan was supported by a NERC Independent Research Fellowship NE/L010992/1. Mark Lunt was supported by NERC grants NE/I027282/1 and NE/M014851/1. Funding for the field campaign was made possible by NERC grant NE/I027282/1.

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

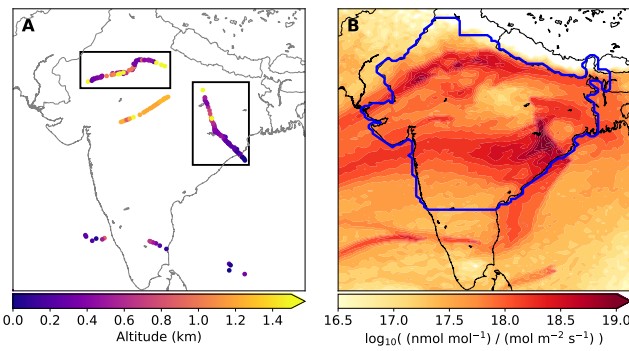

**Figure 1.** (A) Location and altitude of aircraft samples collected over India. The flight paths outlined in boxes were repeated three times each over the sampling period. (B) Average sensitivity to surface emissions from all samples collected over India. A region broadly corresponding to maximum sensitivity in the samples is shown in the blue outline. We denote this region as northern-central India (NCI). The full inversion domain is shown in Fig. S1.

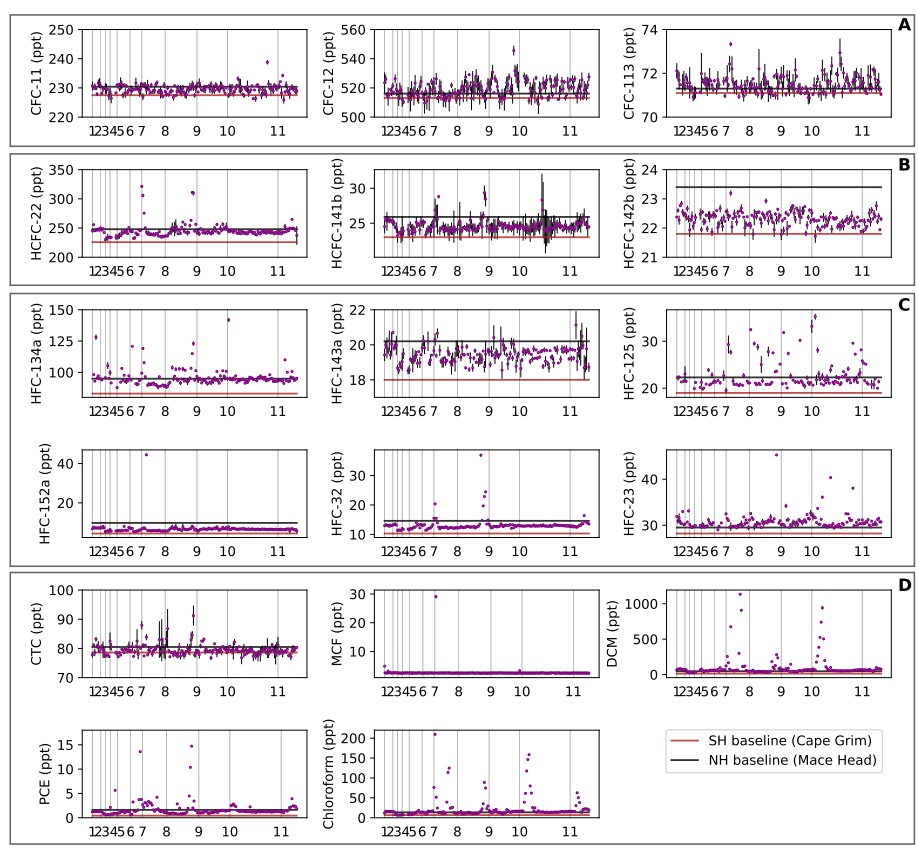

**Figure 2.** (A) CFC, (B) HCFC, (C) HFC and (D) chlorocarbon mole fraction data from 176 flask samples collected over India, plotted on a flight by flight basis (a summary of flights 1-11 is given in Table 2). Error bars represent instrumental precision, which was estimated using the standard deviation of the three replicate analyses of each flask. Two statistical baselines, inferred from observations at Cape Grim, Tasmania (red line) in the Southern Hemisphere (SH) and Mace Head, Ireland (black line) in the Northern Hemisphere (NH), are shown for comparison.

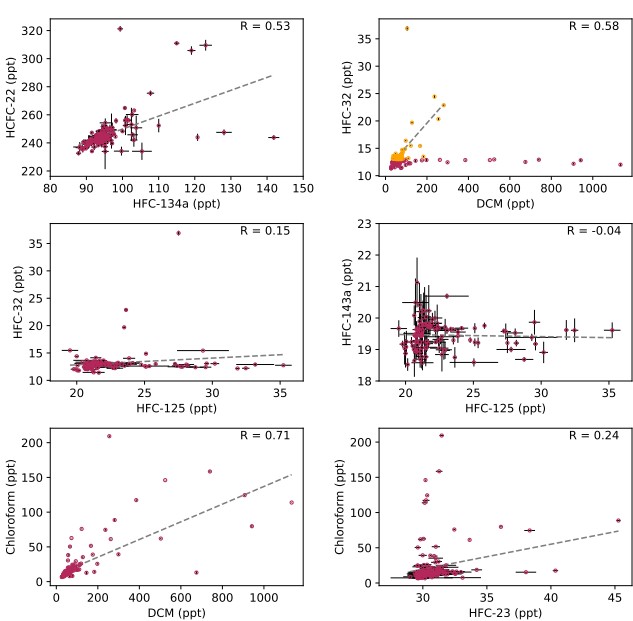

**Figure 3.** Halocarbon scatter plots, shown with line of best fit and Pearson (R) correlation coefficient. For HFC-32 versus DCM, the subscatter shown in orange is a subset of the dataset corresponding to samples whose HFC-32 mole fraction (lower bound of measurement uncertainty) exceeded the $20^{th}$ percentile of all measurements (and were hence classified as enhanced), since there are likely other sources of DCM not linked to HFC-32 production.

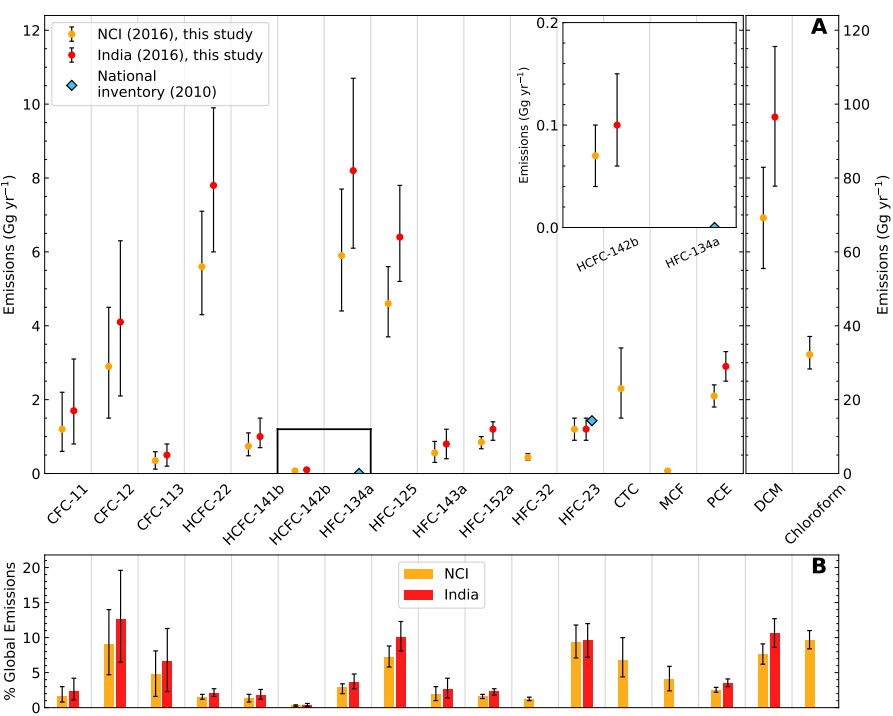

**Figure 4.** (A) NCI (orange) and India total (red) halocarbon emissions (Gg yr$^{-1}$) derived in this study. India's most recent greenhouse gas inventory estimates (2010) are included where available. Note that emissions of DCM and chloroform are presented on a second y-axis for clarity. (B) The estimated contribution of the NCI and India to global halocarbon emissions (global estimates are an extension of the work by Rigby et al. (2014)). Error bars represent the $5^{th}$-$95^{th}$ percentiles of the posterior distribution.

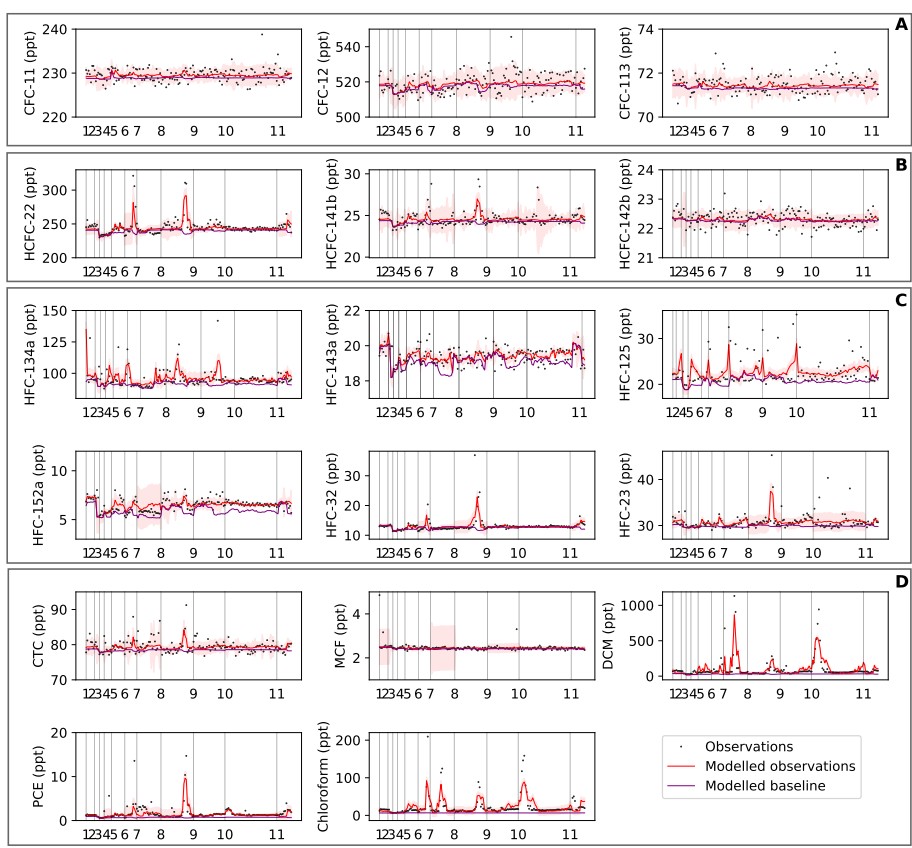

**Figure 5.** Comparison of measured (black points) with posterior modelled (red line) halocarbon mole fraction data, plotted on a flight by flight basis (a summary of flights 1-11 is given in Table 2). The posterior modelled baseline is also shown (purple line). The shading represents the model uncertainty ($5^{th} - 95^{th}$ percentile of the posterior PDF). With the exception of chloroform, for which prior emissions were distributed uniformly over ocean and land, prior emissions were distributed according to the NOAA night light distribution. Note that for HFC-152a and MCF the y-axis has been reduced in comparison to Fig. 2 for clarity.

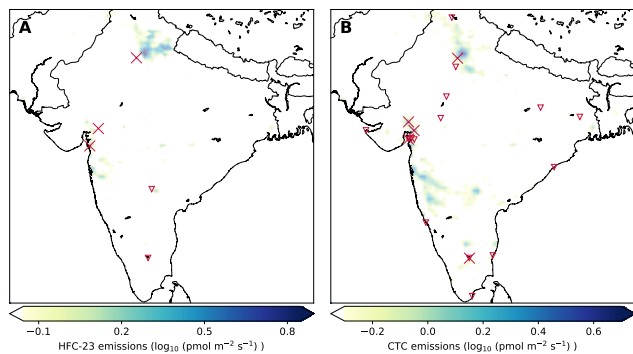

**Figure 6.** (A) Posterior emissions map for HFC-23, reported in pmol m$^{-2}$ s$^{-1}$. The known locations of major (> 8 Gg yr$^{-1}$) and minor (< 1.5 Gg yr$^{-1}$) manufacturers of HCFC-22 are represented by the crosses and open triangles respectively. (B) Posterior emissions map for CTC, reported in pmol m$^{-2}$ s$^{-1}$. The known locations of chloromethane production facilities (crosses) and chlor-alkali plants (open triangles) are also shown.

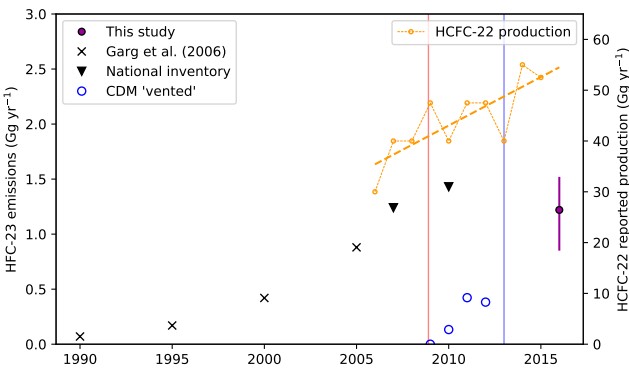

**Figure 7.** HFC-23 emissions (Gg yr$^{-1}$) from bottom-up and top-down estimates. Bottom-up estimates are from Garg et al. (2006) (black crosses) and India's Second National Communication Ministry of Environment, Forest and Climate Change (2012) and Biennial Update Report Ministry of Environment, Forest and Climate Change (2015) to the UNFCCC for 2007 and 2010, respectively (black triangles). The top-down estimate derived here is plotted as a purple circle with corresponding $5^{th}$-$95^{th}$ percentile uncertainties. Blue circles show the total amount of 'vented' (i.e. released to the atmosphere) HFC-23 per year, as reported by the five HCFC-22 manufacturers during the CDM period. Reported HCFC-22 production (Gg yr$^{-1}$) data is shown in orange circles and extrapolated to 2016 using a linear fit (dashed orange line). The red bar indicates the first year (2009) in which all five manufacturers of HCFC-22 reported the use of an abatement system and the blue bar indicates the point (January 2013) at which the European Union banned the use of HFC-23 credits under the EU Emissions Trading Scheme. Note the split y-axes – HFC-23 emissions estimates are plotted with respect to the left-hand axis, while HCFC-22 production data is plotted with respect to the right-hand axis.

**Table 1.** Halocarbons considered in this study. Atmospheric lifetime estimates, ozone-depletion potentials (ODPs) and global warming potentials (100-year time horizon, $GWP_{100}$) are taken from the 2018 Scientific Assessment on Ozone Depletion (Engel et al., 2019). Lifetimes are quoted in years unless otherwise stated.

| Species | Formula | Lifetime | ODP | $GWP_{100}$ | Main application |
|---------|---------|----------|-----|-------------|------------------|
| CFC-11 | $CCl_3F$ | 52 | 1.00 | 5160 | Refrigerant |
| CFC-12 | $CCl_2F_2$ | 102 | 0.77 | 10300 | Refrigerant |
| CFC-113 | $CCl_2FCClF_2$ | 93 | 0.81 | 6080 | Solvent |
| HCFC-22 | $CHClF_2$ | 11.9 | 0.029 | 1780 | Refrigerant |
| HCFC-141b | $CH_3CCl_2F$ | 9.4 | 0.086 | 800 | Foam-blowing |
| HCFC-142b | $CH_3CClF_2$ | 18 | 0.040 | 2070 | Foam-blowing |
| HFC-134a | $CH_2FCF_3$ | 14 | 0 | 1360 | Refrigerant |
| HFC-143a | $CH_3CF_3$ | 51 | 0 | 5080 | Refrigerant |
| HFC-125 | $CHF_2CF_3$ | 30 | 0 | 3450 | Refrigerant |
| HFC-152a | $CH_3CHF_2$ | 1.6 | 0 | 148 | Aerosol propellant |
| HFC-32 | $CH_2F_2$ | 5.4 | 0 | 705 | Refrigerant |
| HFC-23 | $CHF_3$ | 228 | 0 | 12690 | By-product |
| CTC | $CCl_4$ | 32 | 0.89 | 2110 | Cleaning agent |
| MCF | $CH_3CCl_3$ | 5.0 | 0.155 | 153 | Cleaning agent, degreaser |
| DCM | $CH_2Cl_2$ | 180 days | Not well quantified | 10 | Solvent, feedstock |
| PCE | $C_2Cl_4$ | 110 days | Not well quantified | 5.9 | Dry cleaning agent |
| Chloroform | $CHCl_3$ | 183 days | Not well quantified | 18 | Feedstock |

**Table 2.** Aircraft campaign flight summary statistics. IST – Indian Standard Time

| Flight number (Fig. 2 label) | Date (time, IST) | Sampling region | Mean altitude (range, km) | Number of samples |
| --- | --- | --- | --- | --- |
| B957 (1) | 12/06 (06:02 – 07:55) | NE India | 1.20 (0.30 – 7.40) | 9 |
| B959 (2) | 21/06 (08:10 – 08:21) | S India | 0.46 (0.05 – 0.87) | 2 |
| B963 (3) | 25/06 (16:52 – 18:00) | S India | 0.31 (0.21 – 0.53) | 4 |
| B966 (4) | 27/06 (07:12 – 09:49) | S India | 0.30 (0.02 – 0.66) | 9 |
| B968 (5) | 30/06 (05:03 – 06:51) | NW India | 0.98 (0.28 – 3.15) | 11 |
| B969 (6) | 02/07 (05:21 – 07:11) | NW India | 0.53 (0.28 – 0.64) | 11 |
| B971 (7) | 04/07 (07:23 – 08:57) | NE India | 0.38 (0.02 – 1.65) | 20 |
| B972 (8) | 05/07 (05:23 – 07:06) | NW India | 0.83 (0.30 – 1.65) | 27 |
| B974 (9) | 07/07 (06:22 – 07:30) | NW India | 1.29 (0.88 – 2.90) | 26 |
| B975 (10) | 09/07 (06:31 – 08:14) | NE India | 0.37 (0.02 – 1.16) | 44 |
| B976 (11) | 10/07 (06:37 – 07:32) | NW India | 0.42 (0.35 – 0.53) | 12 |

**Table 3.** Halocarbon mass spectrometry target/qualifier ions and respective calibration scales. SIO - Scripps Institution of Oceanography, NOAA - National Oceanic and Atmospheric Administration. Average flask measurement precisions are also shown.

| Species | Target ion ($m/z$) | Qualifier ion ($m/z$) | Calibration scale | Average measurement precision (%) |
|---------|---------|---------|---------|---------|
| CFC-11 | 103 | 105 | SIO-05 | 0.4 |
| CFC-12 | 85 | 87 | SIO-05 | 0.7 |
| CFC-113 | 153 | 155 | SIO-05 | 0.4 |
| HCFC-22 | 67 | 50 | SIO-05 | 0.9 |
| HCFC-141b | 81 | 101 | SIO-05 | 3.4 |
| HCFC-142b | 65 | 85 | SIO-05 | 0.5 |
| HFC-134a | 83 | 33 | SIO-05 | 0.8 |
| HFC-143a | 65 | 64 | SIO-07 | 1.3 |
| HFC-125 | 101 | 51 | SIO-15 | 1.5 |
| HFC-152a | 65 | 46 | SIO-05 | 3.2 |
| HFC-32 | 33 | 51 | SIO-07 | 1.1 |
| HFC-23 | 51 | 69 | SIO-07 | 0.8 |
| CTC | 82 | 84 | SIO-05 | 1.3 |
| MCF | 99 | 97 | SIO-05 | 1.7 |
| DCM | 86 | 84 | SIO-14 | 1.0 |
| PCE | 166 | 164 | NOAA-2003B | 1.0 |
| Chloroform | 83 | 85 | SIO-98 | 0.5 |

**Table 4.** Posterior mean halocarbon emission estimates reported in Gg yr$^{-1}$ for northern-central India and the whole of India, and the percentage contribution of India to global emissions, where appropriate. The $5^{th}$ and $95^{th}$ percentile ranges are shown in parentheses. Asterisks denote that percentages are derived from the NCI total, as scaling to a national total was not considered appropriate for these gases.

| Species | NCI Prior | NCI Posterior | India | % of global |
|---|---|---|---|---|
| CFC-11 | 9.0 | 1.2 (0.6 − 2.2) | 1.7 (0.8 − 3.1) | 2.3 (1.1 − 4.2) |
| CFC-12 | 4.1 | 2.9 (1.5 − 4.5) | 4.1 (2.1 − 6.3) | 12.6 (6.5 − 19.6) |
| CFC-113 | 0.89 | 0.35 (0.12 − 0.59) | 0.49 (0.17 − 0.82) | 6.7 (2.3 − 11.3) |
| HCFC-22 | 8.0 | 5.6 (4.3 − 7.1) | 7.8 (6.0 − 9.9) | 2.1 (1.6 − 2.7) |
| HCFC-141b | 2.1 | 0.7 (0.5 − 1.1) | 1.0 (0.7 − 1.5) | 1.8 (1.2 − 2.6) |
| HCFC-142b | 0.09 | 0.07 (0.04 − 0.10) | 0.10 (0.06 − 0.14) | 0.4 (0.2 − 0.6) |
| HFC-134a | 5.9 | 5.9 (4.4 − 7.7) | 8.2 (6.1 − 10.7) | 3.7 (2.7 − 4.8) |
| HFC-143a | 1.4 | 0.56 (0.30 − 0.87) | 0.8 (0.4 − 1.2) | 2.7 (1.4 − 4.2) |
| HFC-125 | 1.5 | 4.6 (3.7 − 5.6) | 6.4 (5.2 − 7.8) | 10.1 (8.1 − 12.3) |
| HFC-152a | 1.1 | 0.9 (0.7 − 1.0) | 1.2 (0.9 − 1.4) | 2.3 (1.8 − 2.7) |
| HFC-32 | 0.11 | 0.44 (0.36 − 0.54) | - | 1.2 (1.0 - 1.5)* |
| HFC-23 | 1.1 | 1.2 (0.9 − 1.5) | 1.2 (0.9 − 1.5) | 9.6 (7.2 − 12.0) |
| CTC | 2.0 | 2.3 (1.5 - 3.4) | - | 6.8 (4.4 - 10.0)* |
| MCF | 0.2 | 0.07 (0.04 - 0.10) | - | 4.1 (2.4 - 5.9)* |
| DCM | 14.6 | 69.2 (55.5 - 82.9) | 96.5 (77.8 - 115.6) | 10.6 (8.6 - 12.7) |
| PCE | 4.3 | 2.1 (1.8 - 2.4) | 2.9 (2.5 - 3.3) | 3.5 (3.0 - 4.1) |
| Chloroform | 2.2 | 32.2 (28.3 - 37.1) | - | 9.6 (8.4 - 11.0)* |

**Table 5.** Posterior mean halocarbon emission estimates reported in Tg $CO_2$eq $yr^{-1}$ for northern-central India and the whole of India. The $5^{th}$ and $95^{th}$ percentile ranges are shown in parentheses. Emissions totals for the whole of India are not presented for HFC-32, CTC, MCF and chloroform, as scaling to a national total was not considered appropriate for these gases.

| Species | NCI Prior | NCI Posterior | India |
|---|---|---|---|
| CFC-11 | 46.4 | 6.2 (3.1 – 11.4) | 8.8 (4.1 – 16.0) |
| CFC-12 | 42.2 | 29.9 (15.5 - 46.4) | 42.2 (21.6 - 64.9) |
| CFC-113 | 5.4 | 2.1 (0.7 - 3.6) | 3.0 (1.0 - 5.0) |
| HCFC-22 | 14.2 | 10.0 (7.7 - 12.6) | 13.9 (10.7 - 17.6) |
| HCFC-141b | 1.7 | 0.6 (0.4 - 0.9) | 0.8 (0.6 - 1.2) |
| HCFC-142b | 0.19 | 0.14 (0.08 - 0.21) | 0.21 (0.12 - 0.29) |
| HFC-134a | 8.0 | 8.0 (6.0 - 10.5) | 11.2 (8.3 - 14.6) |
| HFC-143a | 7.1 | 2.8 (1.5 - 4.4) | 4.1 (2.0 - 6.1) |
| HFC-125 | 5.2 | 15.9 (12.8 - 19.3) | 22.1 (17.9 - 26.9) |
| HFC-152a | 0.16 | 0.13 (0.10 – 0.15) | 0.18 (0.13 – 0.22) |
| HFC-32 | 0.08 | 0.31 (0.25 – 0.38) | - |
| HFC-23 | 14.0 | 15.2 (11.4 – 19.0) | 15.2 (11.4 – 19.0) |
| CTC | 4.2 | 4.9 (3.2 - 7.2) | - |
| MCF | 0.03 | 0.01 (0.01 - 0.02) | - |
| DCM | 0.15 | 0.69 (0.56 - 0.83) | 0.97 (0.78 - 1.1) |
| PCE | 0.03 | 0.01 (0.01 - 0.01) | 0.02 (0.01 - 0.20) |
| Chloroform | 0.04 | 0.58 (0.51 - 0.67) | - |