# Peer review of "Figure S1. Comparison of wind direction ( $^{\circ}$ ), as measured from the research aircraft (black circles) with those simulated by the NAME model (red circles), for each flight on a minute by minute basis. The points in time at which samples were collected are indicated by vertical purple li"

_Atmospheric Chemistry and Physics, 2018_

## Referee Comment (RC1) · Anonymous Referee #1 · 10 Jan 2019

Review of Emissions of CFC, HCFCs and HFCs from India

General remarks: The paper presents new air plane measurements of halocarbons over India used for the assessment of emissions. I am in favor of publishing the paper after following points have been carefully considered.

Major issues: In the title it could be mentioned that this is based on measurements. Suggestion: Emissions of CFC, HCFCs and HFCs from India based on atmospheric measurements

The introduction and especially the selection of references has been done in a rather careless way. I suggest that the senior authors on this paper could provide some guidance for correc-tion. Under other issues I will mention some but not all of the instances where things should be changed.

[Figure]

other issues: P1. L 3 . . .existing atmospheric measurement networks.

L 6 use km instead of miles

L 13 Our total CFCs. . .

L14 I would delete the second part of the sentence (starting from, suggesting. . .), as this does not mean anything

P2 L2 Wallington seems to be a pretty inappropriate reference for this. Either one of the recent ozone assessments could be cited or Molina and Rowland

L 6 Derwent, Velders, inappropriate: one of the recent ozone assessments would be much better

L9 ODSs

L10 wrong! Emissions have been reduced in the last decade. But they are to some degree reincreasing.

L11 be precise: Montzka as Southeast Asia

L16 Article 5 countries (developing countries). . . remark, nobody outside the Montreal protocol knows that

L17 . . .currently still permitted. . .

P6 L18 not all emissions are on-going so I suggest: . . .of these gases could be ongoing. . .

L20 make a reference to section 2.6, where the model is explained

P8 L1 It is strange here obviously new lifetimes are used but in the table 1 still the outdated lifetimes of Mhyrre are used. The lifetimes from the SPARC report should be used in the table. The GWPs could still be from Mhyrre.

L22-26 This is said again behind. Delete it at one place.

[Figure]

P9 L7 Kim 2009 is nearly a decade old data. This is not recent and this should not be used as a justification at all. Things have changed a lot in China in the last decade.

L24ff I cannot follow the argument here. Why should F-134a increase in the canister? If, then it would decrease and why only F-134a should be affected? I would simply delete this whole argument

P12 L1ff The section about HFC-23 should be under the heading of HFC-23 below.

L26 growth? It can be also a decrease, maybe it is development in India's. . .

P20 Figure 2: looking at the high baseline for HFC-32. Is this reason why the HFC-32 emissions are so low? If so that should definitely be corrected.

P26 Use the SPARC update for lifetimes

P28 Potential mistakes in the table. I hope I saw all but please check. This should not be like that at all!

Potentially wrong: CFC-11 target (T) is it really 103? Not 101? Qualifier (Q) is it really 105, not 103 Potentially wrong: CFC-113 T: 151? Q 153? 141b Q wrong HFC-32 T wrong
* * *

---

## Referee Comment (RC2) · Anonymous Referee #2 · 2 Feb 2019

The manuscript by Say et al., use a series of low altitude airborne measurements of CFCs, HCFCs, and HFCs in combination with models to estimate regional to national emissions of these compounds from India. The analytical measurements were of high quality and provided a useful data set for the emissions modeling. Though I am not really familiar with the modeling techniques used here, the approaches applied seem to be accepted practice and relatively sophisticated for this type of estimation. Overall the manuscript is well written and well organized, and I recommend publication after some revision.

My main concern about the work is the extrapolation to annual emissions of data spanning only several weeks in one limited region of India. The author's assert that the emissions should be reasonably stable over a long period of time, but provide really no evidence that this is true. If emissions are largely from manufacturing, there can be significant variations in emissions from production facilities. Also, as the authors note,

some unexpected seasonality has been observed. There are really no measurements available to check on the various assumptions of conditions during June as being representative of the annual conditions. While error analysis is a significant part of the modeling procedure, there appears to be no estimate of additional uncertainty related to extrapolation of the short and regionally limited data set to annual and national emissions. I would like to see some clearer statement about the overall uncertainty that the authors can ascribe to the national emissions from this extrapolation. Or provide some clear caveat that, "if the emissions calculated for this time period could be scaled uniformly, then the annual emissions would be . . . . . ." Along the same lines, it is unclear to me how uncertainties in the boundary conditions contribute to the final estimate and its uncertainty, and what might be the effect of emission plumes from beyond the Indian borders on the overall estimate of Indian emissions. My understanding is that the boundaries represent some broad regional average from a 12 box model. Would concentrated emissions from Pakistan or East Asia influence the estimates of emissions from India? Further, it was unclear how (if) the Mace Head and Cape Grim measurements were used in the model analysis, or were just used to represent "typical" NH and SH halocarbon levels. My main request for revising the manuscript would be for the authors to more clearly define how their various assumptions contribute to the emissions magnitudes and uncertainties they report.

Some other comments are given below:

P 3 , L 33, Since there may have been some contamination in a few samples, I wonder how long the samples were stored after cleaning and before use on the flights. The note about storage in rooms without air conditioning is relevant for these measurements, but evacuated or even pressurized samples in a container that could get very toasty might also lead to artifacts in canisters with small leaks. P4, L 16, Just because I am curious about statistical calculation, could you describe how you calculated and report the overall standard deviation from triplicate sample measurements? When measurement precisions are shown are these 1 or 2 std deviations? P6, L10-15. These few lines

[Figure]

contain some assumptions that could contribute in some unknown way to the error of the method. As noted, I'd like to have some quantitative estimate of the error. E.g., "climate may minimize this", or "estimates are likely to be representative" or "characteristics are thought to be true". P8, L 16 – 18. Here is where I am not sure about the use of Cape Grim to represent the conditions of the southern model boundary, or the 12-box average. I wonder if the southern boundary (from either source) might overestimate the cleanliness of the regional "unperturbed" Indian background. P9, L1. And Figure 3. While there is some general correlation observed, a correlation coefficient of 0.53 is a weak argument to support common sources. There is significant variability that suggests a variety of different sources (for these and other gases), and significant variability from possibly sporadic point sources. It is this level of variability that causes me concern about extrapolation to the whole year. P9, L 29. I think the author's aren't really talking about stability of HFC-134a, but potential for leakage and artifacts, either before or after sampling (most likely before). P13, L7. I don't think that % of global emissions are expected to scale with just population, so India's 17.7% of world population wouldn't necessarily imply anything about halocarbon emissions. Data availability. I would like to be able to examine the data used in this paper, but I didn't see the data availability and source listed. Title: I agree with the suggestion of the first reviewer to include ". . ...from airborne measurements" in the title.

---

## Author Comment (AC1) · 17 May 2019

**Author responses to peer review of manuscripts '*Emissions of CFCs, HCFCs and HFCs from India*' and '*Atmospheric observations and emissions estimates of ozone-depleting chlorocarbons from India*'**
**Daniel Say on behalf of all co-authors**

We thank the anonymous reviewers for their useful feedback on our manuscripts. Please find below responses to their constructive comments. The merger has resulted in a number of changes to the manuscript – these are shown in the marked-up version where possible. Major changes are listed below.

**AUTHOR NOTES:**
1. Please note that the manuscripts **acp-2018-1146** and **acp-2018-1287** have been merged upon consideration of the comments made by the reviewers of acp-2018-1287. Here follow responses to reviews from both manuscripts, in the order they were submitted. Several of the comments made with respect to **acp-2018-1287** are no longer relevant as a result of the merger – these have been indicated as such.
2. During the review of **acp-2018-1146**, the measurements in Fig. 2 were found to be presented in an incorrect order. This Figure has been replaced by a corrected version - the measurement order is now consistent with those shown in Fig. 5.
3. As a result of the time that has elapsed since submission, the global warming potentials originally quoted are now out of date. These have been replaced in the merged manuscript by those presented in the 2018 Scientific Assessment on Ozone Depletion. Emissions reported in Tg CO2e have been updated accordingly. In all cases the differences are small and do not affect the outcome of the manuscript.
4. Our estimate of India's dichloromethane and chloroform emissions were quoted in error (**acp-2018-1287**). We had mistakenly quoted the NCI total rather than the India total. The correct estimates are 96.5 (77.8 - 115.6) Gg yr-1 (India) and 32.2 (28.3 – 37.1) Gg yr-1 (NCI) - the manuscript has been updated accordingly and the authors apologise for the oversight.
5. In addition to the correlations presented in both **acp-2018-1146** and **acp-2018-1287**, we add the scatter plot of HFC-23 versus chloroform to add further information on the possible sources of chloroform.
6. We remove our previous assertion that the majority of chloroform emissions are from anthropogenic sources. While the correlation with DCM is significant, we do not believe it is strong enough to justify our previous assertion. Since we now believe there to be biogenic emissions of chloroform, we do not scale NCI chloroform emissions to a national total.
7. Our updated paragraph on the production of CTC as a bi-product of chloromethane manufacture now includes discussion alluding to the fact that the difference between our top-down estimate, and the theoretical mass of CTC produced by these facilities, could be due to consumption of CTC in industries such as the production of divinyl acid chloride (P15 L6-8).

**Responses to Review 1, acp-2018-1146**

**Reviewer: Major issues: In the title it could be mentioned that this is based on measurements.**
**Response:** We agree that the title of the manuscript should reflect the fact that the emission estimates are based on atmospheric measurements. Hence, the merged title now reads *'Emissions of halocarbons from India inferred through atmospheric observations'*.

**Reviewer: P1. L 3. Existing atmospheric measurement networks.**
**Response:** Suggestion accepted and added to manuscript (P1 L3).

**Reviewer: L 6. Use km instead of miles.**
**Response:** Suggestion accepted and added to manuscript (P1 L6).

**Reviewer: L 12. Our total CFCs.**
**Response:** Based on the advice of the reviewer, we add the slightly modified '*Our combined CFC estimates*', to ensure that readers appreciate that the proceeding estimates are the combined emissions of CFC-11, CFC-12 and CFC-113 (P1 L13).

**Reviewer: L14. I would delete the second part of the sentence (starting from, suggesting. . .), as this does not mean anything.**
**Response:** Suggestion accepted – *'suggesting that India used a range of HCFC and HFC refrigerants in 2016'* removed from the manuscript.

**Reviewer: P2 L2. Wallington seems to be a pretty inappropriate reference for this. Either one of the recent ozone assessments could be cited or Molina and Rowland.**
**Response:** We accept that a more appropriate reference(s) would be fitting. Hence, we replace Wallington *et al.,* 1994 with Molina and Rowland 1974 and the 2018 Scientific Assessment of Ozone Depletion: Chapter 1 (Engel *et al.,* 2019) on P2 L9.

**Reviewer: L 6. Derwent, Velders, inappropriate: one of the recent ozone assessments would be much better.**
**Response:** We replace Derwent et al., and Velders et al., with the 2018 Scientific Assessment of Ozone Depletion: Chapter 1 (Engel *et al.,* 2014) on P2 L11.

**Reviewer: L 9. ODSs.**
**Response:** Suggestion accepted and manuscript updated accordingly (P2 L11).

**Reviewer: L 10. Wrong! Emissions have been reduced in the last decade. But they are to some degree re-increasing.**
**Response:** We are not entirely clear on the referee comment here, which does say they are re-increasing. Our statement in the manuscript states "*While the emissions of many ODSs are declining, broadly in line with expectations, the emissions of some species, CFC-11 in particular, have not reduced in the last decade, and are now increasing (Montzka et al., 2018).*" Montzka *et al.,* 2018 reported relatively constant global emissions of CFC-11 from 2002 – 2012 (54 ± 3 Gg yr$^{-1}$), followed by an increase of 13.5 ± 5 Gg yr$^{-1}$ to 67 ± 3 Gg yr$^{-1}$ (2014 - 2016). Hence, we do not make any changes to the manuscript with regards to this comment.

**Reviewer: L 11. Be precise: Montzka as Southeast Asia.**
**Response:** Rigby et al., 2019 (Nature) is now in press, and suggests the increase is due to emissions from China. We update the manuscript to state this, P2 L16.

**Reviewer: L 16. Article 5 countries (developing countries) . . . remark, nobody outside the Montreal protocol knows that.**
**Response:** We do not add this definition here because we define developing (Article 5) countries earlier on P2 L12.

**Reviewer: L 17. . .currently still permitted. . .**
**Response:** We replace 'currently' with 'still' on P2 L24.

**Reviewer: P 6 L 18 not all emissions are on-going so I suggest: . . .of these gases could be ongoing. . .**
**Response:** Since we only have evidence of continued emissions for CFC-11, we change *'emissions of these gases are ongoing'* to *'emissions of these gases could be ongoing'* on P7 L23.

**Reviewer: L 20 make a reference to section 2.6, where the model is explained.**
**Response:** Added reference to section 2.6 on P7 L25 – *'(an extension of the work by Rigby et al., 2014, see section 2.6)'*.

**Reviewer: P 8 L 1. It is strange here obviously new lifetimes are used but in the table 1 still the outdated lifetimes of Myhre are used. The lifetimes from the SPARC report should be used in the table. The GWPs could still be from Myhre.**
**Response:** All atmospheric lifetimes are now taken from the 2018 Scientific Assessment on Ozone Depletion (Engel et al., 2019).

**Reviewer: L 22-26. This is said again behind. Delete it at one place.**
**Response:** We remove *'due to the prevailing westerly winds that bring well-mixed oceanic air to the Indian subcontinent during these months. Back trajectory analysis confirmed that these samples had not interacted with any other significant land mass in the 30 days prior to collection.'* in order to prevent repetition of statements.

We also delete *'Except for HFC-134a, the measurements derived from these samples exhibited very little variation, and the mole fractions were amongst the lowest observed during the campaign, which was consistent with the oceanic trajectories. As such, these provided a useful constraint upon the baseline for the modelling studies. In contrast,'* (Originally P9 L 23-25), and add *'Despite this, four of the six…'* on P11 L12.

**Reviewer: P9 L 7. Kim 2010 is nearly a decade old data. This is not recent and this should not be used as a justification at all. Things have changed a lot in China in the last decade.**
**Response:** We agree that Kim *et al.,* 2010 is unlikely to reflect more recent Chinese emission trends. In addition, we also find several other studies reporting a stronger correlation between HFC-125 and HFC-32, which suggests that China does emit (and hence consume) significant quantities of R-410A. Hence, we remove *'In a recent study, Kim et al., 2010 reported a similarly weak relationship for measurements representative of Chinese emissions, suggesting that the two largest Asian economies are yet to adopt the commonly used refrigerant blend R-410A.'* (Originally P9 L7). In its place, we add the lines: *'Conversely, atmospheric measurements from China are consistent with widespread use of R-410A after 2010 (Li et al., 2011, Yao et al., 2012, Wu et al., 2018), perhaps suggesting that India lags behind China in the uptake of the HFC blends designed to replace HCFC-22.'* (P10 L20).

**Reviewer: L 24 I cannot follow the argument here. Why should F-134a increase in the canister? If, then it would decrease and why only F-134a should be affected? I would simply delete this whole argument.**

**Response:** We remove *'Several possible explanations exist for these elevated measurements: 1) Flasks collected over the Arabian Sea were compromised due to long storage times (over 1 month) at temperatures exceeding 40 °C before transport back to the UK for analysis. Long-term tests on the stability of HFC-134a at these temperatures have not been conducted; 2) the enhancements were the result of ship-borne emissions from the Indian Ocean. These flights were at low-altitude (0.01 – 0.8 km) and could have resulted in the measurement of sporadic emissions from ship-based air conditioning systems.'*

In its place we discuss only the possibility that the enhancements are the result of sporadic shipping emissions - *'One possible explanation for enhancements only being observed in HFC-134a over the Arabian Sea is that they are the result of sporadic emissions from ship-based air-conditioning systems, since all Arabian Sea samples were collected at low altitude (0.01 - 0.8 km).'* (P11 L13)

**Reviewer: P12 L 1ff. The section about HFC-23 should be under the heading of HFC-23 below.**
**Response:** We move the paragraph discussing the HFC-23 emissions total (Originally P12 L1-3) to the beginning of subsection 3.3, *'India's HFC-23 emissions and the Clean Development Mechanism'*.

**Reviewer: L 26 growth? It can be also a decrease, maybe it is development in India's. . .**
**Response:** P13 L2 has been re-structured to clarify our point that with a mandate to use an abatement system, India's future HFC-23 emissions may not mirror possible changes in the total volume of HCFC-22 produced. It now reads *'With such systems in place, possible future growth in India's HCFC-22 production rate might not result in increased emissions of HFC-23.'* (P14 L17)

**Reviewer: P20 Figure 2: looking at the high baseline for HFC-32. Is this reason why the HFC-32 emissions are so low? If so that should definitely be corrected.**
**Response:** We believe the reviewer is referring to Figure 5, which shows the derived baseline for each gas in the inversion. The baseline is not high for HFC-32 throughout the period. There is one section between flights 7-8 where the derived baseline is slightly higher than the mean but is still consistent with the uncertainty on the measurements. The majority of the time, the derived baseline in the model matches very well with the baseline measurements. Figure 2 shows HFC-32 measurements alongside data from Mace Head and Cape Grim. The baseline values are between the two sites, as expected (and similar to many of the other gases).

**Reviewer: P26. Use the SPARC update for lifetimes.**
**Response:** All atmospheric lifetimes are now taken from the 2018 Scientific Assessment on Ozone Depletion (Engel et al., 2019).

**Reviewer: P28 Potential mistakes in the table. I hope I saw all but please check. This should not be like that at all! Potentially wrong: CFC-11 target (T) is it really 103? Not 101? Qualifier (Q) is it really 105, not 103 Potentially wrong: CFC-113 T: 151? Q 153? 141b Q wrong HFC-32 T wrong.**
**Response:** We thank the reviewer for pointing out the following errors: HFC-32 target was incorrectly quoted as 23 and has been changed to 33; HCFC-141b was incorrectly quoted as 61 and has been changed to 101. The target and qualifier ions were switched for HFC-23 in order to reflect the correct ion hierarchy. Other *m/z* values are correct and with the exception of HCFC-141b, follow the work by Miller *et al.,* 2008. Upon consideration of similar publications and to simplify the table, we have removed the specific ions, leaving just the *m/z* values for each gas.

**Responses to Review 2, acp-2018-1146**

**Reviewer: My main concern about the work is the extrapolation to annual emissions of data spanning only several weeks in one limited region of India. The author's assert that the emissions should be reasonably stable over a long period of time, but provide really no evidence that this is true. If emissions are largely from manufacturing, there can be significant variations in emissions from production facilities. Also, as the authors note, some unexpected seasonality has been observed. While error analysis is a significant part of the modelling procedure, there appears to be no estimate of additional uncertainty related to extrapolation of the short and regionally limited data set to annual and national emissions. I would like to see some clearer statement about the overall uncertainty that the authors can ascribe to the national emissions from this extrapolation. Or provide some clear caveat that, "if the emissions calculated for this time period could be scaled uniformly, then the annual emissions would be . . . .. ."**

**Response:**
We discuss below (1) the role of production, (2) the extrapolation of emissions estimates from June-July 2016 to an annual average and (3) the extrapolation of emissions from Northern-Central India (NCI) to a national total:

1) In 2016, the only ozone-depleting refrigerant India produced was HCFC-22 (UNDP, 2013). Information from individual manufacturers suggests that HFC-134a and HFC-32 were produced by a single company in 2016 (http://www.srf.com/pdf/media/press/SRF%20Press%20Release_Refigrants02November.pdf). With the exception of HFC-23, whose predominant source is the production of HCFC-22, and HFC-32/CTC and potentially MCF, whose emissions we find are likely to mainly be from production, emissions from production are expected to be significantly smaller than emissions due to consumption for the other gases (Wan et al., 2009, McCulloch et al., 2003). We also discuss in the text on P13 L2 that there could be sporadic sources of HFC-125 in addition to widespread consumption. We therefore make the following changes to the text:

   a. Reports submitted by India's HCFC-22 manufacturers under the Clean Development Mechanism (CDM) suggest that in previous years, production of HCFC-22 (and therefore emissions of HFC-23, assuming immediate venting) did not vary significantly by month. However, there is no such evidence for 2016, and fluctuations in production rate could cause variability in annual HFC-23 emissions not captured in our estimate for June – July. We therefore add the following to P13 L24: *'Emissions of HFC-23 are linked to production of HCFC-22 and could vary in time due to unforeseen facility downtime or fluctuations in demand for HCFC-22. Based on data reported under the CDM (https://cdm.unfccc.int/Projects/registered.html), there is some evidence to suggest that bi-monthly HCFC-22 production rates have, in previous years, remained relatively constant over the course of any given year. However, these reports do not extend to 2016. While the proceeding discussion assumes that our estimate is representative of an annual total, further measurements are required to fully evaluate any short-term variability in emissions of HFC-23.'*

   b. We were unable to find any information regarding the production rate of HFC-32 and therefore add the following caveat regarding HFC-32. To P13 L20 we add: *'In addition, given emissions from production could vary in time (e.g. due to facility down-time), our emissions estimate for this gas should be considered representative of the measurement period.'*

*c.* We add the following (in bold), P7 L16: '*Due to sampling by aircraft, our estimates are likely to be representative on a regional-scale for gases that have sources that are widespread and do not vary significantly in time throughout the measurement period. These characteristics are thought to be true for most gases studied here. With the exception of HFC-23, HFC-32, CTC, MCF and chloroform, emissions **of the other gases** are expected to be dominated by sources linked to consumption **(Wan et al., 2009, McCulloch et al., 2003)**, as opposed to production. **Production could have short-term variations in emissions rate due to, for example, facility down-time. We also discuss below that some caution must be made in the interpretation of HFC-125 emissions.**'*

2) Seasonal variations in emissions rate have been reported for two of the gases discussed in our manuscript, HCFC-22 and HFC-134a. However, in the absence of long-term datasets from the Indian subcontinent, quantification of the magnitude of seasonality is not possible. Xiang et al., 2014 estimated that global emissions of HCFC-22 and HFC-134a are two and three times, respectively, larger in summer than winter due to changes in ambient temperature and air conditioner usage. India's average temperatures do vary by season, with a minimum in winter (January-February, 22.25°C (Indian government statistics)) and a maximum in early Spring (March-May, 28.86°C). In comparison to some of the regions discussed in Xiang et al., (USA, Western Europe), the seasonality is reduced, however we agree with the reviewer the need to discuss this further. We add the following statements:

   **a.** To P7 L3 we add: '*While the estimates presented here represent emissions over a two-month period, they are likely to be consistent with annual emissions for gases that are not expected to have significant seasonality in India. **Seasonal variations in emissions have been observed in HCFC-22 and HFC-134a in Western Europe and North America (Xiang et al., 2014), with summertime emissions that are two and three times larger than wintertime emissions for the two gases. The authors attribute this seasonality to increased vapour pressure in sealed refrigeration/air-conditioning systems as a result of higher ambient temperatures, and to increased use of such systems during summer months. While India's emissions of these gases could exhibit some seasonality, is not possible to estimate the magnitude of this seasonality without long-term observations from the Indian sub-continent. Our estimates for HCFC-22 and HFC-134a should be considered representative of June-July 2016 until long-term studies are conducted. Biogenic sources of chloroform have also been shown to exhibit seasonality (Laturnus et al., 2012), yet emissions from anthropogenic activities (e.g. use as a feedstock) are not likely to vary by season. No such seasonality has been reported for any of the other gases discussed here.***'*

   **b.** To P12 L9 regarding HCFC-22, we add '*Estimating seasonal variations in emission rate for India is not possible without long-term observations. Hence, our estimate for this gas should be considered representative of the measurement period.*'

   **c.** To P12 L24 regarding HFC-134a, we add '*Previous studies reported seasonality in emissions of HFC-134a from Western Europe and North America. Without long-term measurements to quantify this seasonality in India, our emissions rate should be considered representative of the measurement period.*'

3) Quantifying the uncertainty due to scaling emissions from Northern-Central India to a national total by population requires additional measurements from southern India. Several previous studies (e.g. Barletta et al., 2011, Li et al., 2005, Stohl et al., 2009) have used similar methods to scale smaller regions into national totals using population for the gases studied here. In each of these studies, the population of the study area was considerably smaller than the national population. In our study, our NCI domain accounts for approximately 72% of India's total population. Regardless of previous publications, however, we agree with the reviewer that there will be uncertainty when performing any extrapolation.

We now add to P6 L26: '*Emissions were aggregated into totals for the northern-central India (NCI) region (Fig. 1), which contains 72% of India's population, and then extrapolated to a national total for all gases besides HFC-32*, **CTC, MCF and chloroform. The sources of the other gases except HFC-23 are refrigeration, foams, aerosols and landfills, for which we assume population to be a reasonable proxy for scaling emissions, however we are not able to quantify the uncertainty associated with extrapolating to a national total without additional measurements.**'

**Reviewer: Along the same lines, it is unclear to me how uncertainties in the boundary conditions contribute to the final estimate and its uncertainty, and what might be the effect of emission plumes from beyond the Indian borders on the overall estimate of Indian emissions. My understanding is that the boundaries represent some broad regional average from a 12-box model. Would concentrated emissions from Pakistan or East Asia influence the estimates of emissions from India?**

**Response:** The boundary conditions are estimated in the inversion on a domain that is larger than the region for which emissions are presented. We now add to the Supplement (Fig S1), the average sensitivity map over the full NAME domain. We described the domain on P5 L15, but now add reference to Fig S1: 'The model domain spanned from $55 - 109°E$ and $6 - 48°N$ up to 19 kilometres altitude (Fig S1).'

Our inversions are therefore run on a much larger domain than what is shown in Figure 1 (which is curtailed for India for presentation) and therefore includes the effect of emissions from countries outside of India. However, in general these outer regions are sufficiently far from the measurements, that their emissions do not contribute significantly to the mole fraction enhancements over background in Northern-Central India. The NAME sensitivity maps show the significant drop off in sensitivity in these other countries. For example, there is very little sensitivity to East Asia/Pakistan emissions in India at this time of year.

The 12-box model only provides a priori values for the boundary conditions on each horizontal boundary of the full NAME domain. Adjustments to these boundary conditions are then solved for in the inversion to match the 'baseline' mole fractions in the measurements. Any uncertainties in the estimation of the boundary conditions would be absorbed into emissions estimates of the outer regions of the inversion domain by design.

**Reviewer: Further, it was unclear how (if) the Mace Head and Cape Grim measurements were used in the model analysis, or were just used to represent "typical" NH and SH halocarbon levels.**
**Response:** Our paper uses the Mace Head and Cape Grim measurements in Figure 2 to represent typical northern and southern hemisphere baseline mole fractions, providing a useful comparison to our India flask data. These datasets are mainly used visually and are not used directly in the inversion. However, they are used indirectly in that measurements from these sites were used to

derive the modelled semi-hemispheric mole fractions with the AGAGE 12-box model, which were ultimately used to estimate a priori boundary conditions for the inversion (see previous comment).

**Reviewer: P 3, L 33. Since there may have been some contamination in a few samples, I wonder how long the samples were stored after cleaning and before use on the flights. The note about storage in rooms without air conditioning is relevant for these measurements, but evacuated or even pressurized samples in a container that could get very toasty might also lead to artefacts in canisters with small leaks.**
**Response:** Evacuated flasks were stored for up to 2 months prior to filling in the University of Bristol lab, where there is no air conditioning. In addition, the Medusa GCMS instrument measures a wide range of halocarbons and hydrocarbons, and of the anthropogenic species, significant enhancements in the Arabian Sea samples were only observed for HFC-134a. The expectation is that a leak in one or more of the sample flasks would result in enhancements of multiple species. To the best of our knowledge, there have been no comprehensive studies related to the stability of HFC-134a at high temperatures (in stainless steel flasks). However, there are other AGAGE stations that operate in tropical climates. Stainless steel calibration cylinders are sent via non-air-conditioned shipping containers to the AGAGE Barbados site. While the site itself is air-conditioned, this unit has failed on multiple occasions, resulting in lab temperatures that are similar to those in India. Despite this, no issues have been reported for HFC-134a. We therefore think it is more likely that those samples (which all occurred over the Arabian Sea) are picking up ship-based emissions.

**Reviewer: P4, L 16. Just because I am curious about statistical calculation, could you describe how you calculated and report the overall standard deviation from triplicate sample measurements? When measurement precisions are shown are these 1 or 2 std deviations?**
**Response:** The measurement precisions are one standard deviation of the triplicate flask analyses. The sample volume (1.75 L) was reduced in comparison to the analytical set-up described in Miller et al. 2008 (2 L), to allow for triplicate analyses to be conducted.

**Reviewer: P6, L10-15. These few lines contain some assumptions that could contribute in some unknown way to the error of the method. As noted, I'd like to have some quantitative estimate of the error. E.g., "climate may minimize this", or "estimates are likely to be representative" or "characteristics are thought to be true".**
**Response:** See response to comment 1.

**Reviewer: P8, L 16 – 18. Here is where I am not sure about the use of Cape Grim to represent the conditions of the southern model boundary, or the 12-box average. I wonder if the southern boundary (from either source) might overestimate the cleanliness of the regional "unperturbed" Indian background.**
**Response:** The Cape Grim data referred to here (originally P8 L16-18) is used visually to represent a southern hemispheric baseline for reference in Figure 2. Cape Grim data is indirectly used in the 12-box model inversion to estimate a 0-30°S semi-hemisphere value, which is then used as the a priori boundary condition for the southern inversion domain boundary. It is important to note that these a priori values are then adjusted in the inversion because offsets to each boundary are additional parameters in the inversion. They are adjusted to match the mole fractions of the "baseline" data in the measurements. We describe on P9 L4 that '*In addition to emissions parameters, a decomposition of the a priori boundary conditions, represented as offsets to the curtains in the four directions, were also solved for in the inversion.*'

**Reviewer: P9, L1. And Figure 3. While there is some general correlation observed, a correlation coefficient of 0.53 is a weak argument to support common sources. There is significant variability that suggests a variety of different sources (for these and other gases), and significant variability**

**from possibly sporadic point sources. It is this level of variability that causes me concern about extrapolation to the whole year.**
**Response:** An R-value of 0.53 suggests that HCFC-22 and HFC-134a share at least some common sources, or source regions, however, we acknowledge that there could be differences as well. Our work suggests that India is in a transition period, whereby HCFCs are being replaced by HFCs. In 2016, there were still significant emissions of both. However, the rate of uptake of this transition is likely to vary by region and usage, and therefore may not be uniform across all of Northern-Central India. This would contribute to a lower correlation coefficient.

We now add to P10 L11, '*It is likely that these gases share a range of common sources, including use in India's largest refrigeration and air-conditioning sector, stationary air-conditioning (Purohit et al., 2016), **though the rate of transition from HCFC to HFC could vary by region.**'* We also reword P10 L13 so that is now reads '*We find a significant (R = 0.53, Fig. 3) relationship between HFC-134a and HCFC-22 mole fractions, consistent with **some** co-located sources.'*

While there are instances whereby an enhancement in HCFC-22 is not matched by an enhancement in HFC-134a (or vice versa), this does not necessarily mean that the sources of these gases are sporadic in time. For both gases, the model fit is good (Fig 5). Since the model assumes that emissions are constant over the measurement period, a good fit likely means that the emission model (constant emission rate) is able to simulate observations well. An example where this is not the case is HFC-125, and we discuss the fact that there could be due to sporadic sources on P13 L2.

**Reviewer: P9, L 29. I think the author's aren't really talking about stability of HFC-134a, but potential for leakage and artefacts, either before or after sampling (most likely before).**
**Response:** See response to comment 4 for discussion of potential artefacts or instability of HFC-134a at high temperatures.

**Reviewer: P13, L7. I don't think that % of global emissions are expected to scale with just population, so India's 17.7% of world population wouldn't necessarily imply anything about halocarbon emissions.**
**Response:** We had initially included this as a reference point, but we agree that one wouldn't necessarily expect global emissions to distribute according to population because of differences in Article 5 and non-Article 5 nations, production pathways, etc. We now remove any reference to the 17.7% of the global population in the Results and Discussion. However, we continue to use it to create a priori CFC emissions (scaling the global emissions from the 12-box model) as it is the best guess we have. We amend the text on P7 L26, '*To estimate a priori total emissions over India, we scaled an estimate of 2016 global emissions derived using the AGAGE 12-box model (an extension of Rigby et al. (2014)) by population, though CFC emissions are not necessarily expected to distribute globally according to population due to differences in Article 5 versus non-Article 5 country emission trends, amongst other factors.'*

**Reviewer: Data availability. I would like to be able to examine the data used in this paper, but I didn't see the data availability and source listed.**
**Response:** The data has now been uploaded to CEDA, and a link is provided in the data availability section.

**Reviewer: Title: I agree with the suggestion of the first reviewer to include ".....from airborne measurements" in the title.**
**Response:** The title of the manuscript now reads '*Emissions of halocarbons from India inferred through atmospheric measurements'*.

**Responses to Review 1, acp-2018-1287**

**Reviewer: My main comment in the quick report was: What sets this manuscript apart from its companion paper (acp-2018-1146, Emissions of CFCs, HCFCs and HFCs from India). Both report synthetic halocarbon measurements from the same campaign which are even shown to partly correlate with each other due to similar sources.**
**Response:** As per the author notes above, the two manuscripts have now been merged and are presented as a single study.

**Responses to Review 2, acp-2018-1287**

**Reviewer: P 1 Line 8: This has only been 1 month of measurements not 2.**
**Response:** This line was removed upon merging.

**Reviewer: P 2 Line 8: There have been updates to this numbers in Carpenter et al. (2014) and Liang et al. 2018.**
**Response:** This line was removed upon merging. All lifetimes, ODP and GWP values are now taken from the 2018 Scientific Assessment on Ozone Depletion (Engel et al., 2019).

**Reviewer: P 2 Line 17: ODPs**
**Response:** This line was removed upon merging.

**Reviewer: P 2 Line 19: What about the new Chapter 1 of the Ozone Assessment (Engel and Rigby, 2019).**
**Response:** All lifetimes, ODP and GWP values are now taken from the 2018 Scientific Assessment on Ozone Depletion.

**Reviewer: P 2 Line 21 and 22: Hossaini et al and Fang et al is plural therefore, show and estimate.**
**Response:** This line was removed upon merging.

**Reviewer: P5 L25ff. Somehow it is unusual to use different a priori estimates for the individual compounds. Especially questionable in this respect is the use of top-down estimates as an a priori which should be independent of top-down estimates. I suggest that you use the AGAGE-12-box based method for all compounds.**
**Response:** Different methods for compiling prior estimates for each gas were used in order to incorporate the most relevant information for each gas. However, given the large uncertainty assigned to prior in each instance, the absolute magnitude has very little influence on the posterior solution. It is also worth noting that the AGAGE 12-box model is itself a top-down estimate but is based upon independent atmospheric measurements. The main concern raised, which is of lack of independence in the prior, is not an issue here. No prior estimates based on top-down numbers have used any measurements from this study.

**Reviewer: P9 L13 The focus on chloro-alkali plants is a misinterpretation of the literature. It is the total of the production of chlorine related products (chloro-alkane production and chloro-alkali plants). Citation from the conclusion of Hu et al.; Our findings suggest that the majority of US CCl4 emissions could be related to industrial sources associated with chlorine production and processing.**
**Response:** Hu et al., conducted a Bayesian Information Criterion (BIC) analysis to determine the most likely sources of CCl4 in the US. They 'suggest that the distribution of derived posterior emissions is more consistent with that of industrial sources reported by the US EPA TRI (particularly chloralkali production plants)', though we accept that they do not exclude chloromethane production and that this industry is a likely contributor. We therefore modify P15 L16 to include other industrial sources – it now reads '*Ongoing US emissions were attributed to industrial sources, particularly chlor-alkali plants, which differs from our finding that CTC emissions in India do not correspond with known locations of chlor-alkali production.*'

**Reviewer: P9 L16ff What about the correlation of CCl4 with CHCl3. If there is co-production with CH2Cl2, there should also be co-production with CHCl3, please discuss.**
**Response:** The correlation coefficients for DCM vs. CTC and chloroform vs. CTC were both small (i.e., less than 0.2). Chloromethane manufacture is a source of all three of these gases, however, individual plants are likely to produce each component (DCM, chloroform and CTC) at a ratio unique

to that facility. In addition, we now show (P15 L6) that India likely produces more CTC than it emits. Some of this is likely to be consumed by the divinyl acid chloride (DVAC) industry. While we do not know the locations of factories producing DVAC, it is unlikely that all are co-located with chloromethane facilities. Similarly, DCM has a wide range of sources, from solvent use to foam blowing, which are not expected to correlate with the known sources (chloromethane manufacture, DVAC industry) of CTC.

**Reviewer: P11. L13 . . .long-lived chlorocarbons. . .**
**Response:** This line was removed upon merging.

**Reviewer: P22. Table 2. The new Ozone Assessment has the lifetime of CCl4 as 32 years. Please correct and cite accordingly.**
**Response:** All lifetimes, ODP and GWP values are now taken from the 2018 Scientific Assessment on Ozone Depletion, hence the lifetime of CCl4 has been updated to 32 years accordingly.

---

## Author Response (AR2)

**Author responses to reviewer's comments of 'Emissions of halocarbons from India inferred through atmospheric measurements'**

The authors would like to thank the reviewers for their insightful feedback on our merged manuscript, 'Emissions of halocarbons from India inferred through atmospheric measurements'. Please see below a point-by-point response to these comments.

**Editor's comments**

**Editor: Page 01, Line 03: delete existing**
Response: We remove '*existing*' based on the recommendation of the editor.

**Editor: Page 02, Line 04: a concluding statement about the study is missing (e.g. impact of emissions or policy decisions)**
Response: We thank the editor for the suggestion and add the following to the end of the abstract: *'Given the rapid growth of India's economy and the likely increase in demand for halocarbons such as HFCs, the implementation of long-term atmospheric monitoring in the region is urgently required. Our results provide a benchmark, against which future changes to India's halocarbon emissions may be evaluated.'*

**Editor: Page 02, line 21: as a result of**
Response: We replace *'as a consequence of'* with *'as a result of'* based on the recommendation of the reviewer.

**Editor: Page 03, line 07: threat to stratospheric ozone due to**
Response: We add *'ozone'* and apologise for the error. The sentence now reads *'…were not considered a threat to stratospheric ozone due to their short atmospheric lifetimes.'*

**Editor: Page 07, line 04: these gases do not have any seasonal variability?**
Response: We discuss the potential limitations of this assumption on P7 L4-13. Seasonality has been reported for emissions of HCFC-22 and HFC-134a. Given this seasonality, we included caveat statements in the results section for these gases (P12 L16 and P12 L30), stating that our estimates should only be representative of the measurement period. The biogenic component of chloroform emissions is also likely to exhibit significant seasonality. We therefore add a further caveat on P16 L25: *'Likewise, since the biogenic component of these emissions is likely to exhibit significant seasonality, our estimate should only be considered representative of the measurement period.'*

**Editor: Page 10, line 01: what are the characteristics of southern hemispheric distribution then?**
Response: This part of the manuscript focusses on the southern hemispheric mole fraction *baseline*, as opposed to the distribution of emissions. Given the motion of the South Asian monsoon, air drawn up over India during June and July has not interacted with any significant southern hemisphere landmass in at least the 30-days prior to its detection, meaning that it is well-mixed and representative of the hemispheric baseline. Hence, an understanding of the actual distribution of emissions in the southern hemisphere is not required for our discussion.

**Editor: Page 12, line 04: between 2013 and 2016**
Response: We change *'between 2013-2016'* to *'between 2013 and 2016'*.

**Editor: Page 12, line 27: What are the reasons for these discrepancies?**
Response: The reasons for these discrepancies are difficult to ascertain. Like any inventory, India's bottom-up emissions estimates are the subject of considerable uncertainty, which is not well quantified. Without an analysis of the uncertainty associated with the bottom-up estimates, or a detailed description of the methods used to calculate these estimates (which are not available for

India's GHG inventory), we are unable to provide a clear explanation of why these discrepancies exist.

**Editor: Page 15, line 16: between 2008 and 2012**
Response: We change *'between 2008-2012'* to *'between 2008 and 2012'*.

**Editor: Page 16, line 08: what do you mean by the emissions "exported rather than consumed"?**
Response: While there is evidence, both from our measurements and from industrial sources, that India manufactures HFC-32, the lack of correlation between HFC-32 and HFC-125/HFC-143a suggests that India does not consume large quantities of refrigerant blends containing HFC-32. Given the evidence for both manufacture, and lack of emissions, it follows that the majority of HFC-32 manufactured by India is exported. To clarify this, we modify P16 L14 so that it now reads *'Our HFC-32 measurements suggest that a large proportion of the HFC-32 produced is exported, rather than consumed by India itself.'*

**Reviewer 1**
**Reviewer: P1, L13: The following sentence is hard to read: 'Our combined CFC estimates show that India contributed 54 (27 – 86) Tg CO2eq/yr, and HCFC-22 emissions at 7.8 (6.0 – 9.9) Gg/yr were of similar magnitude to emissions of HFC-134a (8.2 (6.1 – 10.7) Gg/yr).' Suggestion: 'Estimated combined CFC emissions for India are 54 (27 – 86) Tg CO2eq/yr. Emissions of HCFC-22 of 7.8 (6.0 – 9.9) Gg/yr were of similar magnitude to emissions of HFC-134a (8.2 (6.1 – 10.7) Gg/yr).'**
Reviewer: We thank the reviewer for the useful suggestion and agree that the sentence was unnecessarily difficult to read. Based on the suggestion of the reviewer, it now reads *'India's combined CFC emissions are estimated to be 54 (27 – 86) Tg CO2eq yr-1. HCFC-22 emissions of 7.8 (6.0 – 9.9) Gg yr-1 are of similar magnitude to emissions of HFC-134a (8.2 (6.1 – 10.7) Gg yr-1)'*.

**Reviewer: P2, L18, Rigby is now published**
Response: We remove *'in press'* from the manuscript and add the complete published citation to the bibliography.

**Reviewer: P11, line 20, why is HFC-32 excluded? It is actually explained further down, so maybe: (excluding HFC-32, see below)**
Response: We add *'see below'* and thank the reviewer for the useful addition.

**Reviewer: P14, L9: The HFC-23/HCFC-22 ratio of 3.1% and the connected discontinuity of the abatement is an important finding. Maybe that could be emphasized in the abstract, in the conclusion or at least in the beginning of the section about HFC-23.**
Response: We agree that the discontinuity of abatement is a key finding of the work. However, we already refer to this finding in the abstract (P1 L14) and conclusion (P17 L14). We do not, therefore, make any additional reference to this finding.

**Reviewer 2**
**Reviewer: P5 Line 8, I think the correct date is 2008 (as indicated elsewhere), not 2012.**
Response: We apologise for the error and replace *'2012'* with *'2008'*.

**Reviewer: I was confused at times with the description of chloromethane manufacture, which was often followed by (i.e., DCM and chloroform). Since they are not talking about CH3Cl (chloromethane), perhaps just leave out "chloromethane" and say only "DCM and chloroform manufacture" or maybe "chloromethanes" manufacture.**
Response: We agree with the reviewer that some clarity is required here and thank them for the 'chloromethanes' suggestion. Throughout the manuscript, we replace *'chloromethane'* with

*'chloromethanes'*, to eliminate any confusion related to the use of chloromethane as a synonym for methyl chloride.

**Reviewer: Table 1. Just to clarify the VSL lifetimes, please include the OH and Temp conditions used to calculate lifetime here.**
Response: The hydroxyl and temperature conditions used to calculate the lifetimes of chloroform, dichloromethane and PCE, can be found in Engel and Rigby et al., (and references within). Given the nature of our paper, we feel these details can be referenced from these studies.

**Reviewer 3**
Reviewer 3 recommended publication as is.

[revised manuscript text omitted]
 are based on the NOAA night lights distribution, but has removed observations where the observed and model simulated wind speed/direction differed by more than 20% removed. Error bars represent the $5^{th} - 95^{th}$ percentiles of the posterior PDF. Note that DCM and chloroform are presented on their own y-axis for clarity.